

# 14 years of lidar measurements of Polar Stratospheric Clouds at the French Antarctic Station Dumont d'Urville

Florent Tencé [1], Julien Jumelet [1], Marie Bouillon [1], David Cugnet [1], Slimane Bekki [1], Sarah Safieddine [1], Philippe Keckhut [1], and Alain Sarkissian [1]

[1]LATMOS, Laboratoire Atmosphères, Milieux, Observations Spatiales, UMR CNRS, IPSL, Sorbonne University/UVSQ, Paris, France

**Correspondence:** Florent Tencé (florent.tence@latmos.ipsl.fr)

**Abstract.** Polar Stratospheric Clouds (PSC) play a critical role in the stratospheric ozone depletion processes. The last 30 years have seen significant improvements in our understanding of the PSC processes but PSC parametrization in global models still remains a challenge, due to the necessary trade-off between the complexity of PSC microphysics and tight model parametrization. The French Antarctic station Dumont d'Urville (DDU, 66.6°S – 140.0°E) has one of the few high latitude ground-based

lidars in the Southern Hemisphere that has been monitoring PSC for decades. This study focuses on the PSC data record during the 2007-2020 period. First, the DDU lidar record is analyzed through three established classification schemes that prove to be mutually consistent: the PSC population observed above DDU is estimated to be of 35% supercooled ternary solutions, more than 55% nitric acid trihydrate mixtures and less than 10% of water-ice dominated PSC. Detailed 2015 lidar measurements are presented to highlight interesting features of PSC fields above DDU. Then, combining a temperature proxy to lidar

measurements, we build a trend of PSC days per year at DDU from ERA5 and NCEP reanalyses fitted on lidar measurements operated at the station. This significant 14-year trend of -5.7 PSC days per decade is consistent with recent temperature satellite measurements at high latitudes. Specific DDU lidar measurements are presented to highlight fine PSC features that are often sub-scale to global models and spaceborne measurements.

## 1   Introduction

Polar Stratospheric Clouds (PSC) have been closely investigated for several decades, primarily due to their critical role in stratospheric ozone chemistry. PSC particles are a combination of water vapour ($H_2O$), nitric acid ($HNO_3$) and sulfuric acid ($H_2SO_4$) in different physical states, and are observed as stacks of layers featuring different mixtures. They form in the winter polar stratosphere when temperature decreases below specific thresholds related to nitric acid and water vapor freezing points. Their main impact is to enable heterogeneous chemical reactions that convert stable chlorine and bromine reservoirs into

active radicals that catalytically deplete ozone in the presence of sunlight (Solomon, 1999). Denitrification and dehydration, mostly through the uptake of $HNO_3$ and $H_2O$ by PSCs as well as PSC sedimentation, subsequently decrease $HNO_3$ and $H_2O$ stratospheric concentration and hence enhance ozone depletion (WMO, 2018). Despite major improvements in the recent years



due to enhanced research and monitoring capabilities, a lot remains to be understood about PSCs, especially on some formation pathways, denitrification and dehydration.

Depending on pressure, temperature, $H_2O$ and $HNO_3$ gas phase concentrations and possibly on the abundance of available nuclei, the three key species combine and form different types of particles and clouds. The nuclei are generally stratospheric sulfur aerosols or to a lesser extent meteoritic material. Water ice crystals (ICE), nitric acid trihydrate (NAT) and supercooled ternary solution droplets (STS) of $H_2O$, $HNO_3$ and $H_2SO_4$ are the three particle types which have been fully characterized in laboratory studies (Carslaw et al., 1997; Hanson and Mauersberger, 1988; Peter and Grooß, 2012). Ice particles can nucleate

both homogeneously and heterogeneously depending on the temperature and air mass history, while NAT particles only nucleate on pre-existing particles. STS droplets form by the uptake of atmospheric gas phase $HNO_3$ by sulfuric acid aerosols. PSC are generally composed of a mixture of these base particles. Depending on their dominant type of particles, different PSCs have different chemical heterogeneous reactivities and therefore different levels of halogen activation. Ice crystals are known to be the most efficient in chlorine activation. However, since they are relatively rare, most of the chlorine activation occurs

on or in liquid particles (Abbatt and Molina, 1992; Hanson and Ravishankara, 1993; Wegner et al., 2012; Tritscher et al., 2021). The efficiencies of STS, NAT, ICE particles and stratospheric aerosols to activate chlorine are compared as a function of temperature in Fig. 1 of Wegner et al. (2012).

    From these three basic particle types and their combinations, more detailed types of PSCs were identified for the purpose of creating classifications based on optical properties of the pure STS, NAT and ICE blends of chemical compounds. The

first was published by Poole and McCormick (1988) in 1988. Since then, several classifications have been proposed, listing different types of clouds based on different set of variables. Over the years, these schemes became more complex as STS, NAT and ICE base types could not explain alone the full extent of observed PSC characteristics and resulted in high proportions of unclassified measurements (up to 30%, Achtert and Tesche (2014), 2018). Additionally, some of the early classification schemes included thermodynamically unstable species at stratospheric conditions which were sometimes detected in laboratory

experiments, notably sulfuric acid tetrahydrate (SAT) or nitric acid dihydrate (NAD) but whose presence was not proven in atmospheric observations. Later, studies showed that PSC often consist of stacks of fine layers of different particle types whose identification depends in the history of the air masses due to hysteresis effects in PSC formation along the temperature scale (Larsen, 2000). Effects of orography both above Arctic and Antarctic peninsulas also lead to considering the temperature cooling rate as an important variable driving orographic PSC formation (Noel and Pitts, 2012).

Overall, the modelling community has not been able to take full advantage of complex PSC classification schemes as most models often have only few variables to resolve PSC microphysics. Therefore, conciliating the complex observational patterns, mostly based on optical properties with a relatively simple PSC parametrization relies on keeping PSC classes as tight as possible, and blending in temperature thresholds to complement observations derived patterns to separate classes with overlapping optical properties. Over time, many PSC classes have been proposed (Browell et al., 1990; Toon et al.,

1990; Stein et al., 1999; Santacesaria et al., 2001; Adriani et al., 2004; Massoli et al., 2006; Blum et al., 2005; Pitts et al., 2009, 2011, 2013, 2018), covering data from ground-based or space-borne lidar measurements. While in-situ measurements with balloons or stratospheric aircrafts are highly valuable, they remain rare. Lidar instruments, both ground-based and space-





borne are more appropriate to study PSCs as they provide extremely high vertical and time resolution data at the cost of a heavy inversion procedure (i.e. as compared to direct particle counters) needed to retrieve the optical properties.

Achtert and Tesche (2014) presented a comprehensive review of the PSC classification schemes. Their study shows that these classifications can lead to very different outcomes when applied to a single dataset. As these classifications are often derived from a single instrument, they are likely to carry their own biases and prevent quantitative comparisons with other datasets carrying other biases related to different instrumental setups. Also, different set of variables have been used to interpret the measurements and the lack of homogeneity in lidar data processing and in the definition of some optical properties make

inter-comparisons between different studies more difficult. Following their conclusions, we decided to consider 3 different classifications proposed by Blum et al. (2005) (2005, hereafter called B05), Pitts et al. (2011) (2011, hereafter called P11), and an updated version of P11, published in 2018 (Pitts et al. (2018), 2018, hereafter called P18) is also considered.

    These classifications are used to analyse the lidar measurements conducted at the French Antarctic station Dumont d'Urville (DDU, 66.6°S – 140.0°E). DDU is located on the shore of the continent and therefore often lies at the edge of the polar

vortex. The station hosts a stratospheric lidar since 1989, making it one of the very few Antarctic station with a long-term lidar data record. The lidar instrument is presented in section 1 along auxiliary satellite and reanalyses data. The lidar data processing as well as the considered classification schemes are then described in section 2. All the results are shown in section 3. First, the outcomes in applying the classification schemes B05, P11 and P18 to the DDU lidar data record are presented and discussed. Second, the analysis of an interesting example of a long lidar measurements session is used to illustrate the

unique capabilities of lidar measurements in characterizing very fine vertical features in PSC fields. Third, lidar measurements and temperature reanalyses from ERA5 and NCEP are combined to produce a PSC occurrence trend from 2007 to 2020. To support the use of the reanalyses temperature, they are compared to temperature radiosondes launched daily at DDU together with temperature measurements from the Infrared Atmospheric Sounding Interferometers (IASI). Finally, some challenges of PSC parametrization in climate models are discussed before exposing the main conclusions.

## 2   Methods

### 2.1   Dumont d'Urville Lidar

Since April 1989, an aerosol/cloud lidar system is in operation at DDU in the framework of the Network for Detection of Atmospheric Composition Changes (NDACC. Originally designed as a PSC monitoring instrument, its capabilities have been extended to study aerosol/cirrus clouds in the Antarctic atmosphere , benefiting of continuous technological upgrades of its

different subsystems. The measurement calendar focuses on the PSC season with nighttime setup, still, the recent focus on aerosol plumes either originating from volcanic or biomass burning activity (Tencé et al., 2022) extended the measurement calendar to the summertime.

    The Rayleigh/Mie/Raman lidar operates at the 532 nm wavelength. $N_2$ vibrational Raman scattering at 607 nm is also acquired. An extensive description of both the instrumental design and inversion procedure is featured in David et al. (2012).

The Nd:YAG laser source emits at 10 Hz frequency with around 250 mJ emitted power in the visible. Backscattered photons



are collected on a collocated 80 cm diameter Newton telescope. A polarizing cube at the reception splits the beam into two components polarized parallel and perpendicular to the laser emission for the 532 nm wavelength, each component is recorded and inverted to gain access to the depolarization ratio. In this study, we use the aerosol depolarisation value defined as the ratio between the perpendicular and parallel particulate backscatter coefficients.

Aerosol vertical profiles will be considered as Backscatter Ratio or Scattering Ratio profiles (hereafter called $R_T$) expressed as the ratio of the total scattering (i.e. including Mie Scattering) to the molecular scattering at a given altitude. Potential saturation effects in the tropopause as well as background noise at mesospheric altitudes are removed from lidar signals. The best time integration window is selected based on the homogeneity of the scene featured on the attenuated $R_T$. Finally, signal inversion is performed using the Klett-Fernald formalism (Klett, 1981, 1985; Fernald, 1984) to derive individual $R_T$ profiles.

The lidar inversion at stratospheric altitude is highly sensitive to the molecular density and the clear-air reference altitude. ERA5 daily meteorological data is used for data processing and clear-air altitude is set between 28 and 32 km according to the signal dynamics.

The total uncertainty on the $R_T$ is estimated to be around 7% on the parallel channel $R_{//}$ up to 28 km, with details available in Tencé et al. (2022). On the perpendicular channel and associated $R_\perp$ backscatter ratio, the altitude-dependent uncertainty

ranges from 10% to 30%. As for the depolarization ratio $\delta(z)$, assuming larger uncertainties on the aerosol depolarization ratio $\delta(z)$ rather than on the linear volume depolarization ratio at relevant altitudes in this paper we estimate the error of around 30% (Tencé et al., 2022).

### 2.1.1   IASI temperature product

The Infrared Atmospheric Sounding Interferometers (IASI) are a series of instruments flying onboard the Metop satellites on

a sun-synchronous orbit. They were launched in 2006 (IASI-A, end of life in 2021), 2012 (IASI-B) and 2018 (IASI-C). Each instruments observes the Earth-atmosphere system with scans of 2200 km. Each scan contains 30 fields of view, each field of view containing 4 pixels. This observation mode allows each IASI instrument to observe every place on Earth twice à day, at 9:30 AM and PM. (Clerbaux et al., 2009).

The IASI instruments are Fourier transform spectrometers that measure spectra of the Earth and atmosphere infrared radiance

between 645 and 2760 $cm^{-1}$ (3.62 and 15.5 $\mu m$) in each pixel. The atmospheric temperature profiles can be retrieved from the radiances observed in the carbon dioxide absorption bands at   700 and   2300 $cm^{-1}$. Bouillon et al. (2022) computed atmospheric temperatures in 10 atmospheric layers between 750 and 7 hPa by first selecting IASI's most sensitive channels to temperatures, and then by using the radiances in these channels as input for an artificial neural network (ANN). The ANN was trained using IASI radiances as input, and the matching ERA5 temperature profiles as output. The validation of the ANN

output against ERA5 and radiosoundings observations showed very good agreement between the three datasets between 750 and 7 hPa. A thirteen-year time series (2008-2020) was constructed with this method, using IASI-A observations from 2008 to 2017 and IASI-B observations from 2018-2020.



## 2.2 Reanalysis products - ERA5 and NCEP

As it will be discussed further in details in the following sections, reanalysis temperature products are often necessary to complement or replace local radiosounding measurements for lidar data processing as well as measurements interpretation. For this study, two reanalysis products are considered: the fifth generation of the reanalysis product of the European Centre for Medium-Range Weather Forecasts (ECMWF), herafter called ERA5, and the reanalysis produced by the National Centers for Environmental Protection (NCEP) and the National Center for Atmospheric Research (NCAR), hereafter called NCEP.

ERA5 is a 4D-Var data assimilation product fully available since January 2019. It is based on the Integrated Forecasting System (IFS) Cy41r2, and provides records of the atmosphere, land surface and ocean waves from 1979 onwards. A detailed description of ERA5 can be found in Hersbach et al. (2020). The temperature product used in this study is gridded on 35 pressure levels (from 975 hPa to 2 hPa), and is 4x daily product, meaning it is available everyday at 00:00, 06:00, 12:00 and 18:00. The product is interpolated at DDU location and the original horizontal resolution is 0.25° x 0.25°. Since ERA5 will be compared with IASI temperature data further in this paper, it should be specified that ERA5 assimilates the measurements of IASI, among other instruments. The interpolation of ERA5 at DDU is also used to calculate the dynamical tropopause, which is the tropopause used throughout this study.

NCEP reanalysis product is the result of a cooperation between NCEP and NCAR. The reanalysis was made available in May 1994 and provides data records from 1957 onwards. Extensive description of the reanalysis is available in Kalnay et al. (1996). This product is available daily at 00:00 UTC and offers the advantage to provide an output for DDU.

## 3 Data processing and classifications

### 3.1 PSC detection by lidar

Beside the lidar data inversion procedure, lidar data pre-processing for PSC consists in dynamic time averaging of the individual 3 minutes raw files, to ensure homogeneity of a lidar scene, either being dominated by clear-air or in aerosol/cloud presence. Fifteen minutes averaged files are first generated, and a preliminary inversion assuming no particular extinction is performed to derive an attenuated scattering ratio. This preliminary product is used to detect the aerosol/cloud presence and tag these 15 minutes files for a next step summation according to homogeneity of atmospheric scenes. This pre-processing minimizes the indirect spatio-temporal smoothing induced by blending clear-air before/after cloud detection and ensures a better Signal to Noise Ratio (SNR).

Once the total backscatter ratio and particle linear depolarization ratio profiles are obtained, a peak detection algorithm is performed on both profiles. Results are combined to produce a set of layers corresponding to the peaks. For each of these layers, the relevant parameters for classifications are computed ($[R_T, R_{//}]$, $[R_\perp, R_{//}]$ and $[R_T, \beta_{tot,\perp}]$) and a type (cirrus, aerosol, or one of the PSC types) is attributed to each layer. Finally, layers that are separated by 300 m or less are merged, which is reasonable given the climatological reality of PSC fields we observe, the lidar vertical resolution of 60 m and the





smoothing applied. Up to five stratospheric layers per profile are retained, sorted according to their total backscatter $R_T$ and
aerosol depolarisation $\delta$ values, which definitions are the following:

$$R_T = \frac{\beta_{aer,//} + \beta_{aer,\perp} + \beta_{mol,//} + \beta_{mol,\perp}}{\beta_{mol,//} + \beta_{mol,\perp}} \tag{1}$$

The parallel and perpendicular backscatter ratio ($R_{//}$ and $R_\perp$), are defined as:

$$R_{//} = \frac{\beta_{aer,//} + \beta_{mol,//}}{\beta_{mol,//}} \tag{2}$$

$$R_\perp = \frac{\beta_{aer,\perp} + \beta_{mol,\perp}}{\beta_{mol,\perp}} \tag{3}$$

The linear particle depolarization ratio is defined as:

$$\delta_{aer} = \frac{\beta_{aer,\perp}}{\beta_{aer,//}} = \frac{R_\perp - 1}{R_{//} - 1}\delta_{mol} \tag{4}$$

where $\delta_{mol} = \frac{\beta_{mol,\perp}}{\beta_{mol,//}}$ is the depolarization ratio of the molecular background and depends on the lidar used. Depending on
the width of the interference filter used on the 532nm channel, some lidars only detect the central Cabannes line, while other
instruments also detect the shifted Raman lines. The interference filter used for DDU lidar has a full width at half maximum
(FWHM) of 1 nm bandwidth and therefore the corresponding molecular depolarization parameter is set to $\delta_{mol}$ = 0.443%
(Behrendt and Nakamura, 2002).

Finally, the total perpendicular backscatter coefficient can be expressed as follows:

$$\beta_{tot,\perp} = \beta_{aer,\perp} + \beta_{mol,\perp} = \beta_{aer,\perp} + \frac{\beta_{mol}}{1 + \frac{1}{\delta_{mol}}} \tag{5}$$

In this paper, $R_T$, $R_{//}$, $R_\perp$, $\delta_{aer}$ and $\beta_{tot,\perp}$ are the relevant lidar related optical parameters considered.

## 3.2    Classification schemes

PSC classification is challenging as described in the introduction, but critical as PSC class directly translates into chemical
efficiency in stratospheric ozone chemistry simulation. Relating thermodynamical or microphysical thresholds to optical ones
is challenging in essence. Literature classifications actually feature different and legitimate thresholds separating some particle
types, some of them we discuss hereafter. The purpose of this study is not to review, rank or assess the relevancy of any
of these classifications especially considering our local and polar vortex-grazing instrumental reference but rather to find a
proper framework to analyse our DDU lidar measurements. Achtert and Tesche (2014) published a comprehensive review of
the existing PSC classifications in the literature. 16 years of PSC measurements at the Swedish arctic station Esrange were





analyzed using seven existing classifications and showed that different schemes resulted in very different outcomes when applied to a single dataset. As a main result, the authors recommend the use of two classifications which show good mutual agreement: the first, B05, is based on Esrange lidar while the second, P11, is built on the space-borne Cloud-Aerosol Lidar with Orthogonal Polarization (CALIOP) on board of the Cloud-Aerosol Lidar and Infrared Pathfinder Satellite Observation (CALIPSO) satellite. The update of P11, P18, was logically included to our study.

B05 is based on the measurements of the ground-based lidar located at Esrange, Sweden conducted between 1996 and 2004. B05 relies on $R_{//}$, $R_{\perp}$, and $\delta_{\mathrm{aer}}$, the latter being related to $R_{//}$ and $R_{\perp}$ via equation 4. This classification sorts PSC measurements into: NAT, STS, ICE and MIX, the latter being a mixed-type one. As for the reference properties, NAT clouds are composed of non-spherical crystals and thus exhibit important depolarization ratio ($\delta_{\mathrm{aer}} > 10\%$) and low parallel backscatter ratio ($R_{//} < 2$). STS PSC are spherical liquid droplets theoretically producing no depolarization and moderate parallel backscatter ratio ($R_{//} < 5$). ICE clouds, due to being dominated by large ice crystals close to the granulometry of cirrus clouds, are associated to depolarization ratios most often largely above 2% and $R_{//}$ between 2 and 7, or only $R_T > 7$. The MIX class theoretically gathers all PSC measurements not fitting the three previously introduced types, even if we just stated above that they cover several different physical or chemical states. B05 types and thresholds are summed up in Fig. 1a.

P11 is based on lidar satellite measurements from CALIOP, operating since 2006. This scheme relies on $R_T$ and $\delta_{\mathrm{aer}}$. While P11 uses these two variables to sort the PSC types it is worth mentioning that $R_T$ and $\beta_{\mathrm{tot},\perp}$ are used to detect PSC layers from the lidar profiles. P11 lists five different types: STS, ICE, NAT-Mix1 (MIX1), NAT-Mix2 (MIX2) and Mixed-enhanced (ENH). ENH particles are most often related to the orographic features that are not expected to be frequent above DDU (Tsias et al., 1999). STS and ICE correspond to the same definitions as in B05, even if the variables used and the corresponding thresholds change. However, B05 MIX category does not correspond to the mixed categories involved here. P11 types and thresholds are summed up in Fig. 1b. Two thresholds are not specified in Fig. 1b: the $\delta_{\mathrm{aer}}$ threshold between ICE and STS, which is a boundary decreasing linearly from $\delta_{\mathrm{aer,STS}} = 0.7\%$ to 0% as $R_T$ increases. The separation between MX1 and MX2 is thoroughly described in Pitts et al. (2009) and comes from optical calculations. This boundary sorts PSC according to their NAT number density or volume: clouds with low NAT number density or low NAT volume will most likely be MX1 while high NAT number density or high NAT volume will be sorted as MX2 (Pitts et al., 2009).

Using spaceborne lidar measurements, a lower signal-to-noise ratio as compared to groundbased measurements is expected, counterbalanced by the sheer amount of profiles acquired over time, once again compared to the groundbased setup. Horizontal averaging as well as a short time integration also contribute to increase noise significance for space-borne measurements. This is the reason why, following the recommendation of Achert and Tesche (2014), some thresholds used in P11 should be adapted when applied to ground-based measurements. The adapted thresholds are shown in Fig. 1b. The RT threshold between background and PSC layers is lowered from 1.25 to 1.1. While Achert and Tesche (2014) advocates the use of $\delta_{\mathrm{aer,STS}} = 0.4\%$ for the STS upper limit of depolarization, we set it to $\delta_{\mathrm{aer,STS}} = 0.7\%$ in order to be consistent with B05, also considering it is a groundbased setup. P11 original value was $\delta_{\mathrm{aer,STS}} = 3.5\%$.

P18 is also based on CALIOP and is the update of P11. It still relies on $R_T$ but now considers $\beta_{\mathrm{tot},\perp}$ instead of $\delta_{\mathrm{aer}}$. As mentioned previously, P11 already used $\beta_{\mathrm{tot},\perp}$ for layers detection. P18 still features STS, ICE and ENH types, but P11 types





MIX1 and MIX2 have been merged in a single category, NAT-mixtures (NATmix). The main update between P11 and P18 is that the latter includes three dynamically computed thresholds. Instead of having fixed values, they depend on the uncertainty

of each measurement or on the abundance of relevant atmospheric species, H2O and HNO3. Such dynamical thresholds are well-adapted to a space-borne lidar which experiences various atmospheric conditions. The three thresholds concerned are $R_{T,thresh}$, $\beta_{tot,\perp,STS}$ and $R_{T,ICE}$. P18 types and thresholds are summed up in Fig. 1c.

In P18, measurement uncertainties are computed with CALIOP-based methods (Liu et al., 2006; Hostetler et al., 2006). When adapting P18 to DDU lidar measurements, we somewhat simplified the dynamical thresholds: $R_{T,thresh}$, the $R_T$ bound-

ary separating background measurements from PSC layers was fixed at $R_{T,thresh}$=1.1 as in P11, following the conclusion of Achtert and Tesche (2014). The threshold $\beta_{tot,\perp,STS}$ separating STS droplets or aerosols from NAT particles and ICE crystals was also fixed, but with a dependence on altitude. The $\beta_{tot,\perp,thresh}$ profile was defined as mean + 1 standard deviation of all $\beta_{tot,\perp}$ values corresponding to background and STS measurements at DDU, per 1 km vertical interval. To do so, B05 and P11 classifications were used. The $\beta_{tot,\perp,thresh}$ profile, not shown here, displays a strong altitude dependency: in the stratospheric

range of PSC occurrences, the threshold variation range spans a whole $10^2$ order or magnitude.

In P18, the $R_{T,ICE}$ value separating NAT mixtures and ENH from ICE is calculated from the nearly coincident H2O and HNO3 MLS measurements, Aura and CALIPSO satellites being both on the A-train constellation. From H2O and HNO3 total abundances, theoretical calculations of $R_T$ lead to a dynamical $R_{T,ICE}$ threshold. The change of $R_T$ NAT/ICE limitation between P11 and P18 is significant: from $R_{T,ICE} = 5$ in P11 to $R_T$ in [2.75 - 4] in P18. Such a change has direct impact on

classification outcomes. We choose not to rely on MLS H2O and HNO3 measurements because coincident measurements are limited and MLS has large uncertainties at PSC altitudes which need to be balanced with large amount of data. We rather set the threshold at $R_{T,ICE} = 3.5$, to be in the range of P18. A greater value would not be adapted to our geographic coastal position. Besides, we tested lower values which lead to a slightly higher fraction of outliers, especially at the tropopause, as we have a different geometry, being ground-based.

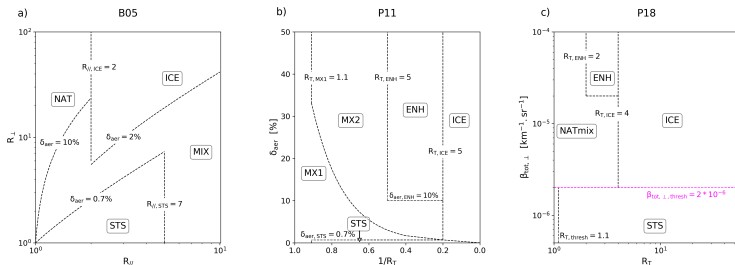

**Figure 1.** The three classification schemes considered in the paper: B05 (a), P11 (b) and P18 (c). The relevant thresholds are specified. The horizontal line separating STS from other PSC types is colored to highlight that the corresponding threshold is not fixed, the given value is only for visualization purposes.



In P11 and P18, the wave ICE category ($R_T > 50$) was ignored since it is not relevant above DDU. ICE PSC induced by orographic gravity waves are very unlikely to be observed at DDU due to its location, and it indeed was never detected.

Most of the published classifications, including B05 and P18, features a mixed-type category. Mixed-type clouds may actually describe different physical realities: a MIX cloud type layer may actually be a fine stack of chemically different layers (finer than the instrumental final vertical resolution of the classification algorithm) or it can also be the signature of particles

outside the thermodynamical equilibrium, and actually evolving along with temperature. The lidar geometry is such that only a small atmospheric air column is probed above the instrument and therefore high variability is expected at stratospheric altitudes with the speed of air masses and different equilibration times for PSC particle population. These disequilibrated particles therefore need to be classified and it is thus expected that they represent a significant part of observations, regardless of the classification scheme.

# 4    Results

## 4.1    Two-variable classification

The PSC types distribution resulting from the application of DDU PSC measurements using B05, P11 and P18 schemes are shown in Fig. 2. A distribution of PSC types at DDU based on CALIOP measurements and P18 scheme published in Tesche et al. (2021) was included to enrich the analysis. Fig. 2 globally shows a very good agreement between the studied classification

schemes. First, the STS abundance is in very good agreement in all cases: between 36.8% and 37.4%. In order to compare the abundance of NAT and NAT mixtures in each classification, the relative abundance of NAT and mixed categories was summed for each scheme, as proper comparison needs to account for the differences in the schemes. For B05, NAT+MIX types weight 56.5%. For P11, MIX1+MIX2+ENH types account for 61.8% of all PSC types. For the schemes based on P18, NATmix+ENH categories make up for 51.2% and 47.3% of the total for the datasets based on DDU and CALIOP measurements respectively.

These small differences reflect the discrepancies between the ICE proportions in all schemes.

The ICE abundance presents an important variability among the four diagrams. The first reason for this variability comes directly from the thresholds considered: as it was previously highlighted, the evolution of $R_{T,ICE}$ from P11 to P18 is significant, from 5 to 2.75, and explains an important discrepancy between the resulting ICE percentages. B05 classifies a PSC as ICE if $\delta_{aer} > 2\%$ and $R_{//}$ in [2,7], or if $R_T > 7$. Furthermore, the actual definition of $R_T$ as $\frac{R_{//} + R_\perp * \delta_{mol}}{1 + \delta_{mol}}$ considering a very low

$\delta_{mol} = 0.00443$ leads to clear dominance of $R_{//}$ in $R_T$. Consequently, the ICE definition in B05 is more permissive than in P11 and P18 to a lesser extent: some PSCs are classified as ICE in B05 while they are considered as NAT in P11 and P18 which is consistent with the differences observed in Figure 2.

The PSC types distribution produced by Tesche et al. (2021) for DDU based on CALIOP measurements and P18 scheme displays 15.8% of relative abundance for the ICE type. As this is based on P18, the previous remark on the role of the ICE

threshold in P18 still applies here. However, the ICE share in Fig. 2d is still significantly higher than in Figure 2b. The difference most likely comes from the point of view of both lidars and different operational constraints. CALIOP, being a space-borne lidar, directly accesses the stratosphere whereas a groundbased instrument sounds the troposphere before reaching it. As for





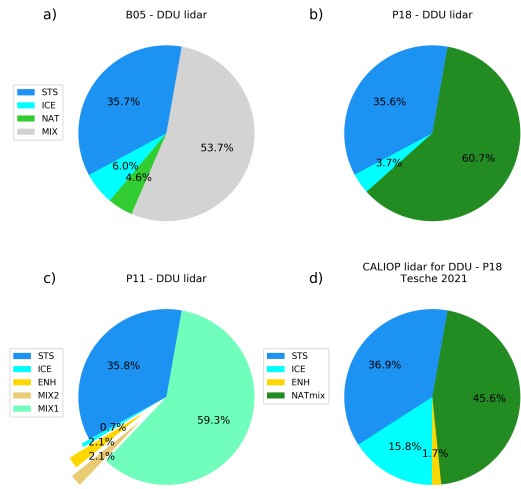

**Figure 2.** PSC types distribution observed at DDU for the three considered classifications: B05 (a), P18 (b), P11 (c) and observed by CALIOP extracted above DDU using P18 scheme (Tesche et al., 2021) (d).

our system, and for instrumental safety concerns, the system is not operated in case of thick tropospheric cloud cover. In fact, studies in the Arctic and the Antarctic suggest a correlation between tropospheric cloudiness and ICE PSC occurrences

(Achtert et al., 2012; Adhikari et al., 2010). Overall, the presence of thick tropospheric clouds is thought to help in reaching PSC-enabling stratospheric temperatures. This reasoning plainly explains the lesser ICE observations above DDU from the ground as compared to CALIOP. Still, ICE PSC occurrences are expected to remain marginal among PSC observations at DDU as the station is located at the edge of the polar vortex and far from orographic gravity waves sources. This small ICE abundance at DDU is supported by Pitts et al. (2018).

The distribution of DDU PSC measurements in B05, P11 and P18 classifications is shown in Fig. 3 and presents where the measurements are concentrated, providing another point of view on the classification thresholds. In Figures 3a and b, it is worth noting that the distribution displays an increase slightly below the STS depolarization threshold. For B05, the absence of measurements along the bottom x-axis is most likely the sign a small crosstalk noise of less than 3% and our algorithm separating STS from aerosol plumes. Then, it appears that most of the measurements are classified as MIX in B05. This is

consistent with Figure 5b from Achtert and Tesche (2014) which also exhibits high densities in the MIX classes. This might be considered as a way to label NAT only the pure NAT clouds, but we tend to consider the 10% depolarization threshold to be too conservative. Considering the prominent share of the NAT class among global PSC observations, we do not explain the small amount of NAT identified by B05.





As for ICE clouds, B05 characterizes significantly more ICE measurements as compared to P11 and P18 where they remain

marginal. The relative low number of ICE events we report has to be combined to optical properties that are also lower at our latitude than would be the ones of clouds observed at higher latitudes. Those optical properties are closer to the respective boundaries of the scheme, and this overall lead to greater variability in characterization among the different schemes. As an illustration, we take the PSC event detected on 2011/07/11 with a depolarization ratio of 3.6% and a total backscatter ratio of 2.3 ($R_{//}$ = 2.034 and $R_{\perp}$ = 9.43). B05 classifies it as ICE while P11 and P18 classify it as MIX1 and NATmix respectively.

From Figs. 3b and c, we conclude that the ENH class is marginal at DDU and is barely never detected.

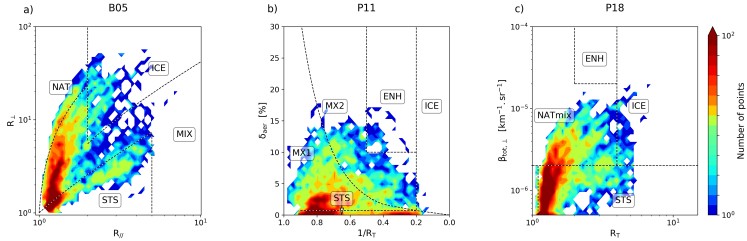

**Figure 3.** Distribution of the PSC measurements acquired at DDU based on the three classification schemes considered: B05 (a), P11 (b) and P18 (c). The number of points per bin is color-coded and each plot is based on a 50x50 bins grid.

The distribution of PSC layers as a function of temperature and altitude was computed for B05, P11 and P18 for STS, NAT+mix and ICE clouds. The category NAT+mix gathers all types related to NAT particles or mixed categories, i.e. NAT + MIX for B05, MIX1+MIX2+ENH for P11 and NATmix for P18 as it is the best way to compare B05, P11 and P18 outcomes. The distributions were computed using kernel density estimation, also referred to as Parzen-Rosenblatt method, a

non-parametric method used to estimate probability density functions of a given sample (Rosenblatt, 1956; Parzen, 1962). The distributions for each classification scheme and PSC type are shown in Fig. 4. The three coloured lines in Fig. 4 indicate the equilibrium temperature of NAT and STS and the frost point in green, brown and purple respectively. These temperatures are called $T_{NAT}$, $T_{STS}$ and $T_{ICE}$ hereafter. $T_{STS}$ relates to a change in the composition of the aerosols and we considered a 50% volume mixing ratio of $HNO_3$ in the condensed phase. Temperature is not a variable in our PSC detection method, yet among

all panels we note that most PSC measurements are below the $T_{NAT}$ threshold.

Figure 4 also shows that the different types have slightly different altitude domains. All classifications agree in that STS mostly form between 15 and 20 km, while ICE are usually detected around the average 20 km altitude. It is important to read the ICE related plots with caution as the densities are computed from a reduced number of points. NAT+mix category occupies a wider domain, from 15 to 25 km. Since this category includes mixtures, it is not surprising that its range is actually wider. As a reminder B05 defines the MIX category as any measurement not belonging to any of the other types. Figures 4g,

h and i illustrate once again the differences in defining the ICE class. The red area between 20 and 25 km in Fig. 4g displays





temperature values too high for ICE and is the consequence of the permissive definition of ICE in B05. P11 classifies few PSC as ICE but those are in the adequate temperature range, while the ones detected by P18 show higher temperatures.

Temperature values associated to PSC types gathered in Fig. 4 are derived from ERA5 analyses and are relevant for the
intercomparison of the classification schemes but they may not resolve small scale variations or features that are important for PSC formations pathways. The relevancy of ERA5 reanalyses in the study of PSC at DDU is reviewed in sect. 3.3. Moreover, the threshold temperatures are computed based on climatological values of $HNO_3$ and $H_2O$ stratospheric concentrations (10 ppb and 5 ppm, respectively) which are generally accurate in the beginning of winter but decrease as dehydration and denitrification take place.

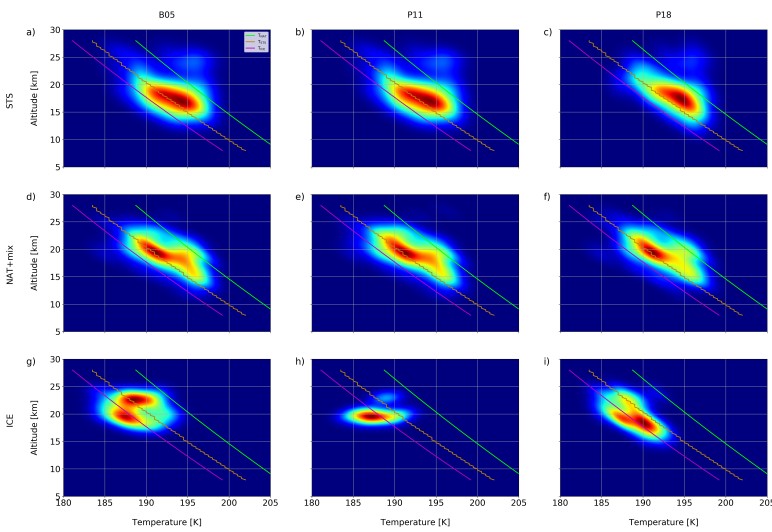

**Figure 4.** Distribution of PSC detection at DDU from 2007 to 2020 as a function of temperature and altitude. The temperature thresholds $T_{NAT}$, $T_{STS}$ and $T_{ICE}$ are shown in green, brown and purple lines respectively. Lines correspond to STS, NAT + mixtures and ICE from top to bottom. Columns correspond to the classification schemes B05, P11 and P18 from left to right.

**4.2 Sample PSC event of interest: 2015/08/28**

Figure 4 showed that B05, P11 and P18 have comparable outcomes. The most recent scheme (P18), which also features the least amount of classes, was therefore selected to report a DDU PSC event as illustration of the detection methodology. The lidar time series is presented in Fig. 5a. At the cost of a reduced SNR implied by the 15-minute time integration, the short-scale dynamics of the PSC layers are visible. The signal is horizontally smoothed on a 30-minute window. The measurements shown
in Fig. 5 are obtained on August 2015 the 28th from 10:05 AM until 08:20 PM. As elements of context, we present in Figure A2 the outputs of a chemistry-transport model available in the institute (REPROBUS coupled to the MIMOSA transport scheme),





resolving PSC formation from the thermodynamical equilibrium assumption and a modal size scheme for the microphysics, accounting for temperature tracers of time elapsed below $T_{NAT}$ and $T_{ICE}$. References on the transport model can be found in Hauchecorne et al. (2002) and on the chemistry module in (Lefèvre et al., 1994). Figure A2 shows PSC flag presence at the

435, 475 and 550K potential temperature levels computed from ERA5 reanalyses. Our measurements fully validate the model in that PSC presence is detected at the 435 K and 475 K levels, and no PSC at the 550 K level above DDU.

Figure 5a also highlights the high temporal variability of PSC layers at the DDU latitude. This high variability must be kept in mind as well as the trade-off on the integration time between SNR and horizontal smoothing (i.e. information loss) due to the transport. Fig. 5 also underlines the contrast between the reality of a complex shape of the 3D PSC field and the necessary

and legitimate stance of recent classification schemes to keep things as simple as possible. PSC are evolving depending on their stratospheric environment. While PSC particles are constantly growing or shrinking, taking up $H_2O$ or $HNO_3$ from the gas phase or enriching it, the classification schemes keep up with fixed thresholds as temperature history cannot be accounted for. On Fig. 5b around 5PM, the type identified for the PSC located at 20 km changes several times from NATmix to ICE. Since this measurement is processed with P18, it implies that RT has just crossed the RICE threshold. The PSC chemical composition

may not have changed, but its optical properties may have shifted it from one class to another.

Figures 5a and b also illustrate the stack of fine PSC layers, it is especially clear between 10 AM and 3 PM. In the beginning of the measurement session, some of the layers at the bottom of the stratosphere are interpreted as STS while the upper ones are classified as NATmix. Then, starting from noon and until the end of the session, $R_T$ increases and the PSC becomes ice dominated. This evolution is consistent with the ERA5 temperatures shown in Fig. 5c. In the first half of the session the

temperature of the domain between 10 and 18 km is around $T_{ICE}$ but then falls below this threshold on a thinner domain in agreement with the lidar PSC detection of Figs. 5a and b which layer becomes thinner as time elapses during the day.

To further check the temperature evolution regarding $T_{ICE}$, Fig. 6a compares $T_{ICE}$ (sky blue dashed line) to ERA5 temperatures profiles at 11 AM, 6 PM and 11 PM (yellow, green and black lines respectively). ERA5 profile at 11PM is included in order to be compared to the radiosonde temperature launched at DDU at 11 PM (blue line). The discrepancies between

ERA5 and DDU radiosonde temperatures are actually associated to the drift of the radiosonde as its altitude increases. Figs. 6b and c show ERA5 $T - T_{ICE}$ fields around DDU at 11PM at 70 hPa and 100 hPa (approximately 15 and 17 km respectively) together with the trajectory of DDU radiosonde (black and red dots). Red dots show the location of the radiosonde between 14 and 18 km, i.e. the boundaries of the ICE PSC detected at DDU. This one example highlights the recurrent drift of the radiosonde leading to use of reanalyses for climatological purposes. While ERA5 temperature between 14 and 18 km at DDU

are compatible with an ICE PSC, the radiosonde tells a different story. Radiosonde drift is closely connected to temperature uncertainties, and we discuss this point in the next section, as temperature is always a critical variable in threshold processes like cloud formation.

### 4.3 Temperature datasets

Lidar data processing heavily relies on accurate temperature data due to Rayleigh scattering and extinction correction to get

the Mie contribution out of the raw signal. PSC characterization is also closely connected to temperature thresholds, as shown



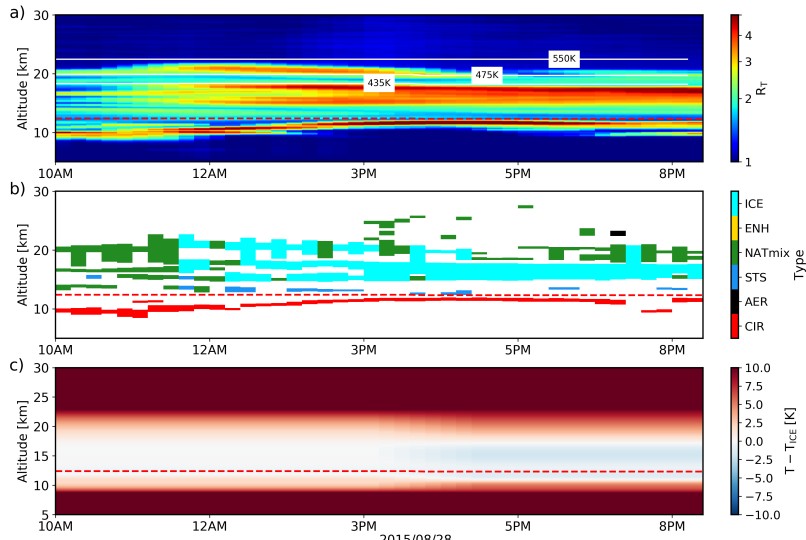

**Figure 5.** 532 nm backscatter ratio of lidar measurements obtained at DDU on the 2015/08/28 (a). The corresponding PSC types according to P18 classification scheme (b) and ERA5 temperatures at DDU as compared to the ICE formation threshold $T_{ICE}$ (c). The red dashed line indicates the dynamical tropopause computed from ERA5 data.

in Figs. 5 and 6. In that regard, the choice of temperature dataset should be done with caution. In order to investigate the effect of temperature variation on PSCs, we use in Fig. 7 different temperature datasets, from reanalysis to satellite observations. Fig. 7a shows the distribution of DDU radiosondes locations between 15 and 25 km, i.e. the stratospheric altitude range of most PSC occurrences, from 2010 to 2020 (radiosondes were not equipped with a GPS before 2010). Figures 7b, c and d

present the temperature difference between ERA5 (Fig. 7b - $\Delta T_{ERA5-RS}$), NCEP (Fig. 7c - $\Delta T_{NCEP-RS}$) and IASI (Fig. 7d - $\Delta T_{IASI-RS}$) with respect to DDU radiosondes from June to September, from 2007 to 2020. The IASI temperature product provides high spatial resolution and daily temperature profiles included in a narrow box of $0.6°$ longitude width and $0.3°$ latitude height centred on DDU from 2008 onwards. The area used to extract IASI temperature profiles is delimited by a red dashed rectangle in Fig. 7a.

Temperature profiles from DDU radiosondes were not retained for this climatological study for two reasons. First, they are launched at DDU every day around 9 AM local time, and the lidar is operated at nighttime, so a discrepancy might arise, especially during summer. Second, as it can be inferred from Figs. 7b, c and d, radiosondes often burst reaching between 15 and 25 km in the wintertime (see Fig. A2 in Annex) and therefore do not provide the full temperature vertical profile. Figure 7a highlights this important horizontal transport. Figure A2 in Annex shows the distribution of the distance to DDU at which

radiosondes burst, from 2010 to 2020, in summer and in winter. To illustrate that such horizontal transport can have significant



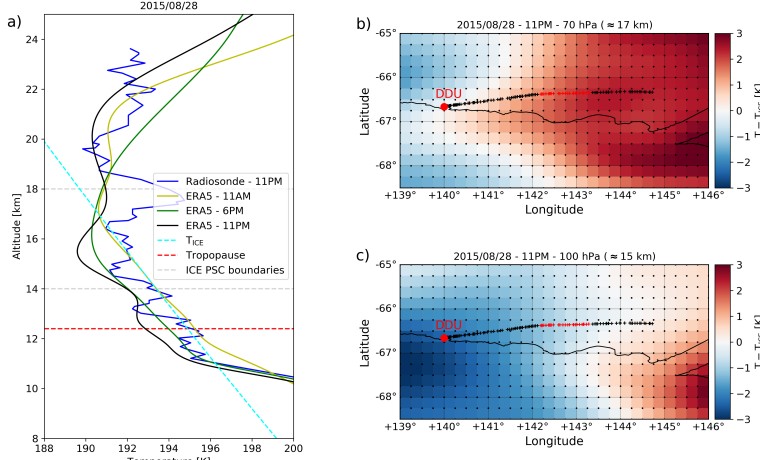

**Figure 6.** (a) 2015/08/28 DDU temperature profiles from a local radiosonde launched at 11 PM (blue line), ERA5 reanalyses at 12 AM (green line) and 6 PM (yellow line). $T_{ICE}$ threshold temperature is indicated in sky blue dashed line, the red dashed line shows the dynamical tropopause from ERA5. The boundaries of the ICE PSC detected at DDU from approximately 3 PM to 8 PM are shown in grey dashed lines. (b) and (c) Temperature fields from ERA5 reanalyses on 2015/08/28 at 11 PM respectively at 70 hPa, approx. 17 km, and 100 hPa, approx. 15 km. The trajectory of the radiosonde launched at DDU on 2015/08/28 at 11 PM is shown in black and red dots. Red dots correspond to heights between 14 and 18 km i.e. the boundaries of the ICE PSC detected at DDU.

impact on the temperature retrieved by radiosondes as compared to the stratospheric conditions above DDU, Fig. A3 in Annex presents six examples of ERA5 temperature fields over the same geographical areas as Fig. 7a. These examples show the variety of spatial temperature patterns experienced around DDU and emphasize that radiosondes should be used with caution as this important horizontal transport is often not taken into account. At least during polar winter, temperature profiles retrieved
from radiosondes should not be considered as purely representative of the launch pad location.

We basically consider three temperature datasets, two are reanalyses, ERA5 and NCEP, and the third one is an inversion product out of IASI satellite radiances. Figures 7b, c and d present an intercomparison with radiosondes launched at DDU from June to September, from 2007 to 2020. First, NCEP is obviously less accurate that ERA5 and IASI, both in the troposphere and in the stratosphere. The difference between ERA5 and IASI seems not to be significant and the comparison with radiosondes exhibits altitude-dependent patterns. For ERA5, $\Delta T_{ERA5-RS}$ is positive in the stratosphere and negative in the
upper troposphere. For IASI, a positive $\Delta T_{IASI-RS}$ bias is recorded below the tropopause suggesting a difficulty in assessing the tropopause height, or a vertical resolution problem (the IASI dataset is a 11-layer product between 750 and 7 hPa, whereas ERA5 has a finer resolution. To quantify the deviation of ERA5, NCEP and IASI with respect to the radiosondes, the standard




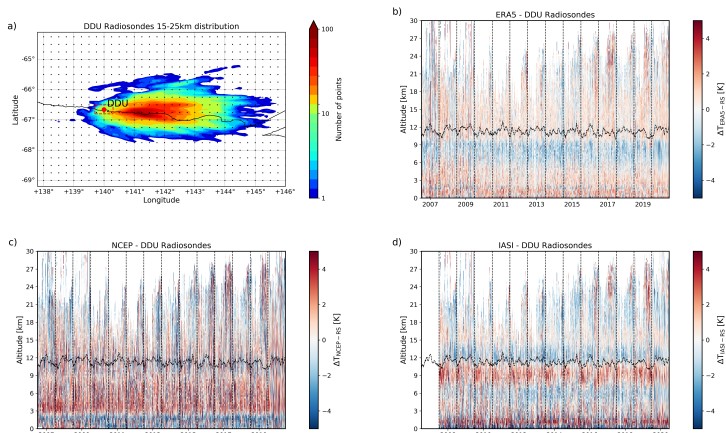

**Figure 7.** Spatial distribution of radiosondes measurements between 15 and 25 km from June to September from 2010 to 2020 (a). The number of points per bin is color-coded, the grid size is 100x100 bins. Black dots indicate the ERA5 grid. The red dashed rectangle delimits the area used to extract IASI temperature profiles. Difference between temperature given by ERA5 (a), NCEP (b) and measured by IASI (c) as compared to radiosondes launched at DDU from June to September from 2007 to 2020. The black dashed line indicates the dynamic tropopause based on ERA5.

deviation of $\Delta T_{\mathrm{ERA5-RS}}$, $\Delta T_{\mathrm{NCEP-RS}}$ and $\Delta T_{\mathrm{IASI-RS}}$ was computed between 15 and 25 km. It reaches 1.0 K for ERA5 and 2.0 K for NCEP and 1.1 K for IASI. Considering the above statements, and the finer vertical resolution of the ERA5 temperature product, we consider the use of the ERA5 temperature the most relevant to our study.

## 4.4 PSC trend estimation

The two major roles of a ground station are to perform process studies and establish decadal trends. Such trends are highly valuable because they reflect the evolution of the stratosphere, in terms of temperature and chemical compositions. Operating instruments at high latitudes for decades remains a technical and logistical challenge, and the focus is put on operational capabilities. Bad weather or thick tropospheric cloud cover hinders the operation of the lidar. Therefore, comparing the raw number of PSC days per year would be strongly biased by the number of days on which the lidar is effectively operated each year. The statistics of operations is the critical point in establishing a trend here. We choose to complement the statistics of lidar measurements with a temperature proxy, considering PSC form when temperature drops below $T_{\mathrm{NAT}}$. For qualitative and counting purposes, this assumption is fully valid and will be illustrated hereafter. Both NCEP and ERA5 reanalyses are used to compile stratospheric temperature above DDU. Using these reanalyses, a number of potential PSC days is computed, i.e. the number of days where PSCs could occur based only on temperature. The number of PSC days per year is the variable considered in order to circumvent the challenging issue of delimiting PSC both in time and space.





Mainly due to chemical kinetic concerns, PSC generally form a few degrees below $T_{NAT}$ (Dye et al., 1992). Using the
condition $T - T_{NAT} < 0$ to state if a day is a potential PSC day could then lead to an overestimation of the number of PSCs.
In order to refine the criterion, we computed each year the number of days satisfying the condition $T - T_{NAT} < \Delta T$ per year,
with $\Delta T$ ranging from 0 K to -10 K on the days lidar measurements were available. The results obtained with the different
$\Delta T$ values were compared with the number of PSC days detected by the lidar. The result lead to a similar value for both ERA5
and NCEP of $\Delta T = -2K$. Figure 8 shows the 2007-2020 PSC days per year built on ERA5 (green line) and NCEP (blue line)
based on this criteria, as well as the number of PSC days detected by the lidar with red triangles. The grey arrows indicate the
number of days per year satisfying the $T - T_{NAT} < -2K$ criteria with no coincident lidar measurements.

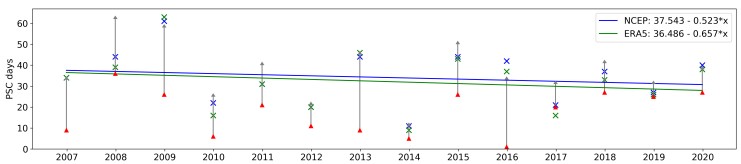

**Figure 8.** PSC days per year at DDU from 2007 to 2020 featuring PSC detection with the lidar in red triangles. Potential PSC days per year
estimated by ERA5 and NCEP based on the lidar measurements are shown in green and red respectively. Green and blue lines represent the
corresponding trends. Grey arrows indicate the number of days per year where the $T - T_{NAT} < -2K$ criterion was satisfied and DDU lidar
was not operated.

The amount of PSC detected by the lidar in Fig. 8 is consistently and logically below the one estimated by NCEP and
ERA5. From Fig. 8, we note a decreasing trend of -5.7 PSC days per decade that could not have been inferred from the lidar
measurements alone. The 14-year trend remains significant when its sensitivity is tested regarding the $\Delta$ criterion or the impact
of any single year. Given the temperature-based criterion used, this trend means that the temperature of the stratosphere above
DDU is experiencing an opposite trend, as PSC occurrences directly correlate to temperature changes.

Temperature trends computed with IASI over the period 2008-2020 period show significant warming above Dumont d'Urville
(0.1 K/year between 200 and 70 hPa, adapted from Bouillon et al., 2022, shown in Fig. A4). ERA5 trends over the same period
show very similar results. As mentioned in Tritscher et al. (2021), the acceleration of the Brewer-Dobson circulation linked to
climate change counteracts the expected cooling from greenhouse gases at high latitudes. The recovery of the ozone hole also
advocates for a rise of polar stratospheric temperatures (WMO, 2018).

If this 14-year trend is negative, it is not the case for longer time spans: using the same method on the period 1992-2020, we
conclude no significant trend. This is consistent with recent studies concluding on no significant trend of PSC occurrences on
the continental scale using longer time periods (Tritscher et al., 2021). It is worth noting that David et al. (2010) also established
a local temperature trend using 50 years of balloon radiosoundings above DDU, concluding on no significant temperature trend
during winter. David et al. (2010) also concludes on a positive but statistically not significant PSC occurrences frequency trend
over the period 1989-2008. Similarly, trends computed with ERA5 over 1990-2020 show smaller or insignificant warming

(Bouillon, 2021), which is coherent with the results of others studies where trends were computed on a longer period (Randel et al., 2016; Maycock et al., 2018). This might suggest that the ozone hole recovery has been strengthening in the past decade.

Discussing PSC estimation by models, Tritscher et al. (2021) stated that the PSC volumes derived from ERA-Interim using the $T < T_{NAT}$ criteria are about 50% overestimated as compared to the satellite PSC measurements. Such an overestimation may be due to reanalysis uncertainties especially in the calculation of $T_{NAT}$. Mostly, this PSC volume evaluation assumes that PSC layers entirely fill the available stratospheric volume satisfying $T < T_{NAT}$. This hypothesis does not seem in agreement with our observations at DDU. To check this above DDU, the total stratospheric range filled by PSC layers is computed and

the corresponding stratospheric range satisfying $T < T_{NAT}$ is calculated with ERA5. The distribution of both PSC thickness and $T - T_{NAT}$ stratospheric domain in presence of a PSC are shown in Figure 9. In Fig. 9, the PSC days with a $T < T_{NAT}$ range of 0km read as days where DDU lidar detects PSC but ERA5 reanalysis does not indicate stratospheric temperature below $T_{NAT}$. Apart from this point, Fig. 9 tends to show that the stratospheric domain satisfying $T < T_{NAT}$ is significantly larger than the actual range filled by PSC layers and seems roughly in agreement with the 50% overestimation indicated by

Tritscher et al. (2021). Given the fact that ERA5 slightly overestimates stratospheric temperature at DDU according to Fig. 7b, the discrepancy between PSC thickness and the stratospheric domain satisfying $T < T_{NAT}$ could even be underestimated.

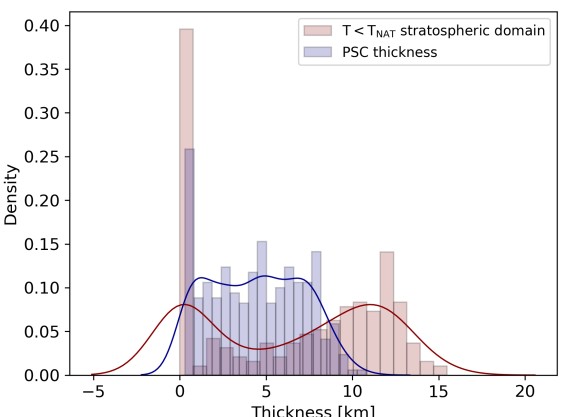

**Figure 9.** Distribution of the total PSC thickness per day in km (blue) and of the stratospheric domain satisfying $T < T_{NAT}$ when a PSC is detected at DDU.

## 5   Conclusions

The stratospheric DDU Rayleigh/Mie/Raman lidar is one of the few instruments monitoring stratospheric aerosol and clouds activity in Antarctica for decades. This study presents PSC measurements acquired from 2007 to 2020. The high vertical and

spectral characterization capabilities of lidar instruments remain the best suited to characterize any given particle population,





especially at stratospheric altitudes. Optical properties overall shape PSC classifications, which in turn drive the parametrizations of PSC in models. Over time, many PSC schemes were published using different species, mixtures, optical variables, and separation thresholds. In this paper we analyse DDU PSC measurements using 3 major schemes referred to as B05, P11 and P18, the first relies on ground based measurements and the other two on CALIOP spaceborne measurements. Laying our

measurements on these schemes, a good mutual agreement between all three is established. ICE cloud ratio still vary from one scheme to another but these differences are directly explained by the design of the classifications. DDU measurements are also compared to a PSC types distribution at DDU based on CALIOP PSC measurements published by Tesche et al. (2021). Ground-based and space-borne measurements agree relatively well on the types distribution, showing significant disagreement only for the ICE type, but the relatively low number of observations at the DDU latitude explains most of the variability. Con-

nected to this, an established correlation between ICE cloud formation and tropospheric cloud cover should also explain the variability, as it prevents our instrument to operate. The spaceborne geometry indeed has always direct access to stratospheric altitude without any significant particle extinction. Correlating PSC formation temperature to the three chemical classes, ICE clouds are observed at a higher altitude than STS and NAT clouds.

  From typical lidar time series, we highlight the small-scale features of PSC layers as well as their temporal variability

both in vertical extent and optical properties, discussing time integration influence (both from a ground and space geometry) when using thresholds to characterize cloud types. While PSC schemes are often built on a massive amount of data, smaller local datasets are able to support class definition by refining mesoscale observed behaviours. We also extensively consider the sensitivity of temperature datasets as small scale proxies for PSC formation thresholds. Overall, we emphasize the local variability of the measurements acquired at the station closely related to the dynamics of the vortex and prevalence of horizontal

transport at stratospheric altitudes. We compare ERA5 and NCEP reanalyses to a IASI-derived temperature product and also to local radiosondes. Related to the coastal location of the station, a significant spatial drift of the sondes during their ascent up to the stratosphere lead us to use temperature reanalyses in lidar data processing to ensure consistency. ERA5 proves to be the most convenient dataset to use locally, showing a satisfying agreement with DDU radiosondes from 2007 to 2020. In the near future, the IASI product should become an interesting option, especially for high latitude sites: the accuracy seems to be

the same as ERA5, except in the upper troposphere.

  A temperature proxy statistically complements DDU lidar PSC measurement days and we build a 14-year trend of number of PSC days per year. A significant slightly negative (-5.7 PSC days / decade) trend is found between 2007 and 2020 and relates to an opposite trend in term of stratospheric temperatures for southern high latitudes which was also reported in a recent study (Bouillon et al., 2022).

PSC volume is often estimated in models as the stratospheric volume satisfying $T < T_{NAT}$, with acknowledged overestimations in derived volumes up to 50% (Tritscher et al., 2021). DDU lidar measurements show that the PSC detected are often significantly thinner that the stratospheric domain satisfying $T < T_{NAT}$. This should have an impact on chemical efficiencies of chemical compounds conversion rates involved in stratospheric ozone chemistry but it is beyond the scope of our paper.

  Finally, DDU offers a privileged access to air mass entries into the vortex, which has been studied recently after the 2019

Australian wildfire event above DDU (Tencé et al., 2022). The global impact of volcanic or biomass burning aerosols through





long range transport now attracts more scientific attention, and these events feature, especially after months of transport, optical properties that overlap the one of some PSC types, mainly STS. Speciation between stratospheric sulfate, carbonaceous aerosols and STS PSC type requires extensive measurement capabilities in monitoring stations. Delving into any potential interplay between PSC and aerosol layers also demands consolidated PSC classification schemes.

*Data availability.* The Dumont d'Urville lidar instrument is part of the NDACC international network and data are publicly available online at the NDACC/NOAA data archive (https://ftp.cpc.ncep.noaa.gov/ndacc/ncep/). ERA5 reanalysis data are available on the Copernicus Climate Data Store at https://cds.climate.copernicus.eu/. NCEP / NCAR reanalysis data are available at https://ftp.cpc.ncep.noaa.gov/ndacc/ncep/.





# Appendix

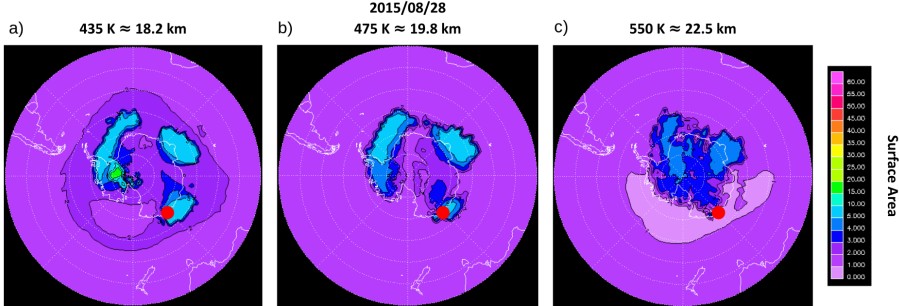

**Figure A1.** PSC surface aera [$\mu m^2/cm^3$] maps from Reprobus model for the 2015/08/28 at 435 K (a), 475 K (b) and 550 K (c). DDU location is indicated by the red dots.

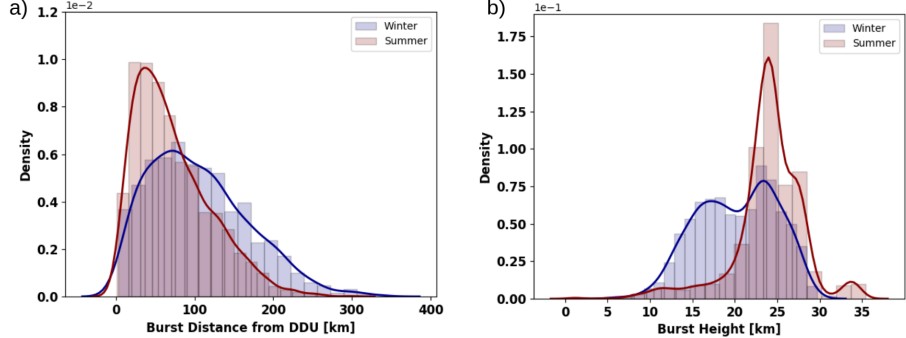

**Figure A2.** (a) Distribution of the distance between DDU and the burst location of DDU radiosondes, in summer (red) and winter (blue) from 2010 to 2020. (b) Distribution of the burst height, in summer (red) and winter (blue) from 2010 to 2020. Winter is defined as the period from June to September included, and Summer is the rest of the year.

*Author contributions.* FT and JJ designed methodology and the core of the paper, and processed lidar measurements. MB and SS produced and provided the IASI temperature product. DC computed ERA5 temperature product for DDU. SB, PK and AS supervised the project and provided expertise.

*Competing interests.* The contact author has declared that neither they nor their co-authors have any competing interests.



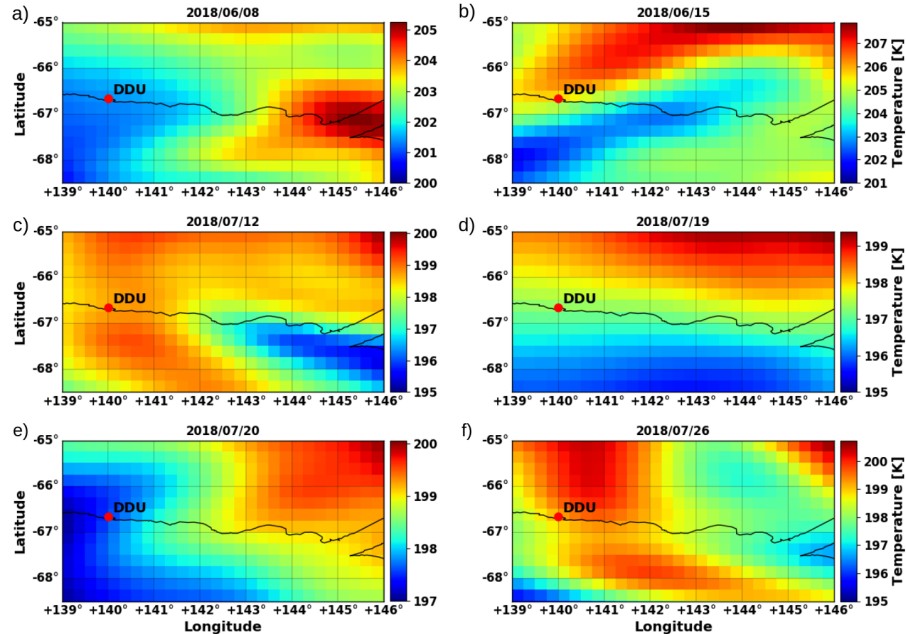

**Figure A3.** ERA5 reanalyses temperature fields at 100 hPa for the domain corresponding to the radiosondes drift shown in Figure 7.

*Acknowledgements.* Operations on the Dumont d'Urville station are supported by the French Polar Institute IPEV (Institut polaire français
Paul-Emile Victor), science program 209. This work was also supported within the EECLAT project (CNES-INSU). The lidar instrument is
part of the NDACC international network.





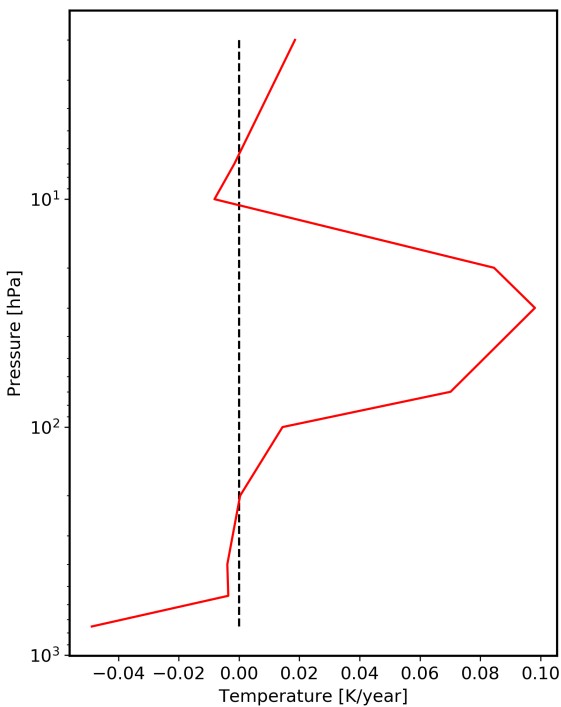

**Figure A4.** IASI Temperature trend at DDU, from 2008 to 2020. Adapted from (Bouillon et al., 2022)

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
