# Peer review of "years of lidar measurements of Polar Stratospheric Clouds at the French Antarctic Station Dumont d'Urville"

_Atmospheric Chemistry and Physics, 2022_

## Referee Comment (RC2)

Review of acp-2022-40: **14 years of lidar measurements of Polar Stratospheric Clouds at the French Antarctic Station Dumont d'Urville,** by Tencé et al.

This paper presents analyses of a 14-year database of PSC observations from the ground-based lidar at the French Antarctic station Dumont d'Urville (DDU). After describing the lidar system at DDU, the authors present an extensive evaluation of three published PSC classification schemes they consider for use with the DDU database. One of the classification schemes is then selected to use to illustrate a sample PSC event of interest. Next, the authors evaluate the viability of several temperature datasets (radiosondes, ERA5, NCEP, and IASI) for use in analysis of the DDU PSC data. Their analyses indicate that the ERA5 temperature data is best suited for use in processing and interpretation of the DDU PSC data. Finally, the authors derive a PSC trend estimation utilizing temperature statistics as a surrogate for PSC data due to sampling issues in the ground-based lidar database. Their results show a decreasing trend of -5.7 PSC days per decade based on 14 years of data (2007-2020).

The overall goals of the paper are not clearly articulated in the abstract or introduction and remain unclear to this reviewer. A significant portion (almost 50%) of the paper is dedicated to the evaluation of the three published PSC composition classification schemes, B05 (Blum et al., 2005), P11 (Pitts et al., 2011) and P18 (Pitts et al., 2018). Although the authors state that the purpose of their study is "not to review, rank or assess the relevancy of these classifications … but rather to find a proper framework to analyse our DDU lidar measurements," it seems that reviewing and assessing the classifications is indeed a primary focus of the paper. After the extensive discussion of the three classification schemes, the authors conclude that the three are comparable, but select the P18 scheme to examine a sample PSC event of interest. If a detailed comparison of the schemes is not a focus of the paper (and I don't think it should be), then it would be more succinct to simply describe the classification approach the authors think most appropriate and then present more detailed analyses of the 14-year PSC database using this classification. What new insights into PSCs can be gleaned from the 14-year DDU database? For instance, instead of showing just one sample PSC event of interest, the authors could present some multi-year statistics on the mesoscale characteristics of PSCs. This could be a unique contribution from the DDU dataset.

The discussion of the temperature datasets could also be much more succinct providing the justification for why the ERA5 data is chosen to be used in the DDU data processing and analyses. Again- depends on what are the primary goals of the paper.

The trend analysis is interesting, but the inclusion of the CALIOP data here (see detailed comments below) could provide additional insight.

As I'm sure the authors are aware, there are at least two other recent published studies of PSCs using ground-based lidars located in the Antarctic:

Snels et al. (2021). Quasi-coincident observations of polar stratospheric clouds by ground-based lidar and CALIOP at Concordia (Dome C, Antarctica) from 2014 to 2018, Atmospheric Chemistry and Physics, 21, 2165–2178.

Snels et al. (2019). Comparison of Antarctic polar stratospheric cloud observations by ground-based and space-borne lidar and relevance for chemistry-climate models. Atmospheric Chemistry and Physics, 19, 955–972.

How do the DDU PSC statistics compare with these other datasets? I would expect the papers to at least be cited somewhere in this manuscript.

Then finally, there are numerous instances of awkward and/or confusing text throughout the manuscript. I have pointed out many of these below and suggest some wording changes. Although individually they are generally not major issues, the large number of these instances make the manuscript difficult to follow and should be revised where appropriate.

In conclusion, although I do have issues with many portions of the manuscript, I believe with some significant revisions and possible inclusion of new analyses (as described below), it may be suitable for publication in ACP.

**Specific Comments:**

1) The current Introduction in Section 1 needs major reworking. What are the main goals of this study? How are the ancillary datasets used? These need to be clearly articulated here. This will provide a roadmap for the rest of the paper. From the current introduction, it's not clear why so much effort is going into evaluating the classification schemes. Is this a major goal of the paper? The last paragraph of the current Introduction is attempting to describe the remaining sections of the paper- but it is not accurate or complete. Please rewrite to better summarize what is in each subsequent section (e.g., 2.-Methods, 3- Lidar data processing, 4- Results, 5-Conclusions).

2) Line 3: The meaning of the term "tight model parameterization" is not clear. Do you mean mathematically simple and/or computationally fast?

3) Line 17: What is meant by "stacks of layers featuring different mixtures." Please cite a reference and describe in a little more detail what is meant by this phrase.

4) Line 18: change "when temperature" to "when the temperature"

5) Lines 20-22: Strictly speaking, I believe denitrification and dehydration refer to the redistribution and **irreversible** removal of $HNO_3$ and $H_2O$ from the stratosphere. Uptake of $HNO_3$ and $H_2O$ by itself (through particle formation) may be reversible. Therefore, denitrification and dehydration occur by sedimentation of large NAT or ice PSC particles that contain $HNO_3$ and/or $H_2O$.

6) Line 23: The phrase "a lot" is not a good choice for a technical paper. Would be good to have some citations here on what significant improvements have taken place and what remains to be understood. Perhaps more relevant here, is this study going to improve our understanding of any of these outstanding questions?

7) Line 27: "sulfur aerosols" should be "sulfuric acid aerosols"

8) Line 27: meteoritic material- is there a citation you could include here that shows meteoritic material may be efficient PSC nuclei?

9) Lines 27-29: I suggest listing relevant citations in the same order as the particle compositions (ice, NAT, STS) … (Peter and Grooß, 2012; Hanson and Mauersberger, 1988; Carslaw et al., 1997). Did Peter and Grooß (2012) actually perform lab studies on ice? I thought this was a chapter in a book.

10) Lines 30-31: "NAT particles only nucleate on pre-existing particles." - what pre-existing particles? ice? meteoritic material? Citations?

11) Line 37: The Wegner et al. (2012) Figure 1 only shows efficiencies for liquid aerosol (binary and ternary) and NAT, not ice. Aren't these efficiencies primarily based on the available surface area? Is it really composition dependent or mostly surface area density dependent?

12) Lines 38-39: This sentence is confusing to me. What do you mean by "pure STS, NAT, and ICE blends of chemical compounds?" Are you simply referring to the chemical makeup of the particles?

13) Line 40: "Poole and McCormick (1988) in 1988." I think it is obvious that Poole and McCormick was published in 1988, so you don't need the additional "in 1988"

14) Line 41: "set" should be plural "sets"

15) Line 43: "Achtert and Tesche (2014), 2018)." I think this is a typo- ", 2018)" should be deleted.

16) Line 45-46: "… but whose presence was not proven in atmospheric observations." Suggest rephrasing as "but has yet to be confirmed by atmospheric observations."

17) Lines 46-48: What studies have shown these "stacks of fine layers?" I don't think that the Larsen paper shows that PSCs often occur as stacks of layers of different particle types- at least I didn't see any mention of this in the report. Larsen does conclude that the temperature history of the air mass must be known to properly simulate the particle formation.

18) Lines 48-49: Sentence is poorly worded. Suggest something like "In addition, the temperature cooling rate is an important variable driving orographic PSC formation in both the Arctic and Antarctic (Noel and Pitts, 2012)."

19) Line 51: "only few" should be "only a few"

20) Lines 51-54: This sentence is not clear and too long. What is based on "optical properties"? The complex observational patterns? Surely not the parameterization schemes? Numerous phrases that are not clear: "tight as possible"? "observations derived patterns"? Please try to reword.

21) Lines 54-56: This sentence seems repetitive with the sentence L.40-42. Maybe you can combine this with the sentence on L. 40-42 and list the citations there?

22) Line 63: "different set of" should be "different sets of"

23) Lines 65-67: Years inside the parentheses are not necessary. Suggest rewording "… Blum et al. (2005) (hereafter called B05), Pitts et al. (2011) (hereafter called P11), and Pitts et al. (2018) (an updated version of P11, hereafter called P18)."

24) Consideration of P11: The P18 algorithm corrected several know deficiencies in the P11 algorithm. I believe the P18 has replaced P11 as the operational algorithm used to produce the CALIOP v2 PSC data products. Therefore, there is no reason to include the P11 version in your evaluation unless you just want to compare the differences between P11 and P18 (that was done by Pitts et al., 2018). Is that your goal here?

25) Line 70: suggest changing "station hosts" to "station has hosted"

26) Summary paragraph beginning on Line 68: As mentioned above, this last paragraph in the Introduction needs completely rewritten. This paragraph is attempting to describe the following sections of the paper- but is not accurate or complete. Please rewrite to better summarize what is in each subsequent section.

27) Line 80: Section 2 Methods: This section really doesn't describe methods- rather just the datasets used in the study. Probably should rename "Datasets"?

28) Line 82: Suggest changing "Since April 1989, an aerosol/cloud lidar system is in operation at DDU …" to "An aerosol/cloud lidar system has been in operation at DDU since April 1989 …"

29) Line 83: Add closing parenthesis after NDACC

30) Line 84: Delete extra space after "Antarctic atmosphere"

31) Lines 85-87: Awkward grammar- suggest rewording this sentence to "Although the measurement calendar focuses on the PSC season with nighttime setup, the recent focus on aerosol plumes either originating from volcanic or biomass burning activity (Tencé et al., 2022) has extended the measurement calendar to the summertime."

32) Section 2.1 DDU Lidar description: You say that the lidar capabilities have been continuously upgraded and cite the David et al. (2012) paper. Have there been any notable upgrades in the past 10 years since the David et al. paper?

33) Section 2.1 DDU Lidar description: Since you are introducing most if not all of the lidar optical parameters here, I suggest you move the equations defining the lidar parameters in Section 3.1 to this section. It seems more appropriate to have the definitions here. Maybe after L. 94?

34) Line 97: "saturation effects" - Would you please describe what the saturation effects are and add more detail on how they are removed?

35) Line 98: "homogeneity of the scene" –How is the homogeneity quantified and used?

36) Line 100: "altitude" should be plural "altitudes"

37) Section 2.1.1 IASI temperature product: What is the main role of the IASI temperature product in this study? Should note that in the introduction (section 1).

38) Line 111: "instruments" should be singular "instrument"

39) Line 112: à should be "a"

40) Line 113: delete period after PM

41) Line 118: "temperatures" should be "temperature"

42) Line 120: "very good agreement" – Please be more quantitative- what is very good agreement?

43) Lines 124-125: Sentence is worded awkwardly. Suggest rewording something like "As discussed in further detail in the following sections, reanalysis temperature products are

often utilized to complement or replace local radiosonde measurements for both data processing and interpretation of ground-based lidar measurements."

44) Lines 129-130: What do these acronyms (4D-Var, Cy41r2) mean?

45) Line 133: "interpolated at DDU location" – "interpolated to the DDU location" Is it simply linearly interpolated from the original product grid (0.25 x 0.25 degree)?

46) Line 135: "dynamic tropopause"- how do you define the dynamic tropopause? Is this an ERA5 product that you interpolate to the DDU location?

47) Line 139: NCEP product: What is meant by "provide an output for DDU"?

48) Reanlyses Data discussion in general: Again, there should be some discussion in the Introduction of how the reanalyses data will be used in this study. What are the uncertainties in reanalyses data products?

49) Section 3.1 PSC detection by lidar: I found the description of the PSC detection here to be confusing. The first step is some "pre-processing" that identifies time segments that contain aerosol/cloud? What is dynamic time averaging? What do you mean by "next step summation according to homogeneity"? What is the "peak detection algorithm"? Are you just searching each profile identified as containing aerosol/cloud for peaks that identify layers? The output of the detection algorithm are profiles of lidar parameters with one or more layers identified as being aerosol/cloud? You mention in Lines 150-152 that a type is attributed to each layer. Isn't the composition classification performed separately from detection and dependent on the specific scheme being used as described below in Section 3.2? Please try to rewrite this section more clearly with more detail.

50) Line 144: "assuming no particular extinction" – do you mean "assuming no particulate extinction"?

51) Lines 155-169: As mentioned above, I suggest you move these lidar parameter definitions up to Section 2.1

52) Section 3.2 Classification schemes: General comment- I think too much emphasis has been put on discussion and evaluation of the classification schemes. You state that the purpose of the study is not to "review, rank or assess" the classification schemes, but to "find a proper framework" to analyse your data. It sure seems that you are reviewing and assessing the classifications. What does "proper framework" mean? How do you decide which classification scheme provides the proper framework.

53) Lines 176-182: Much of this was already discussed in the Introduction.

54) Line 225: "$10^2$ order or magnitude" should be "$10^2$ order of magnitude" Do you really mean 100 orders of magnitude or just 2 orders of magnitude (factor of 100 in magnitude)?

55) Lines 230-232: I agree that it doesn't make sense to use the MLS measurements directly. But wouldn't be better to use a climatology of HNO3 and H2O and have a time dependent threshold? It would be straight forward to produce a climatology from the MLS data.

56) Line 235: Suggest rewording this to read: "The wave ice category defined in P11 and P18 was ignored in this study ..."

57) Line 236: verb tense doesn't match- "… published classifications … features" should be "…published classifications … feature"

58) Line 242: Suggest changing "disequilibrated" to "non-equilibrium"

59) Discussion of Figure 2: Figure 2d (derived from Tesche et al., 2021) is based on only two Antarctic seasons (2012 and 2015) and I believe uses only a subset of CALIOP measurements randomly selected to represent the possible sampling of a ground-based lidar that is affected by cloudiness and other measurement-inhibiting factors. What is the relevance of this figure to the others in Figure 2 that are based on 14 years of data? Doesn't seem to be a fair comparison. It certainly would be straight forward to derive a new figure using the CALIOP data for the same timeframe as your DDU data- then the comparison would be more meaningful.

60) Discussion of ice discrepancy in Lines 271-274: To better investigate this- I suggest you subset the CALIOP data to the DDU location and evaluate the ICE abundance on days when the DDU lidar operated versus days in which DDU lidar didn't operate.

61) Lines 280-283: These two sentences are not clear- not sure what you're trying to say.

62) Lines 285-286: Why would optical properties be dependent on latitude?

63) Line 290: "barely never" – that phrase makes no sense. Do you mean "barely ever"? Probably would be better to just say "rarely detected."

64) Figure 4 and corresponding discussion: Using threshold temperature values calculated with fixed values of HNO3 (10 ppbv) and H2O (5 ppmv) will likely produce misleading results. These values may be appropriate for early season (at ~50 hPa), but clearly are not representative for the bulk of the season after denitrification and dehydration have occurred. In reality, the gas phase abundances are much lower over most of the season and the threshold temperatures will correspondingly be lower. It would not be too difficult to derive a climatology of HNO3 and H2O from MLS data that reflects the seasonal and altitude variation and then use this to calculate time dependent temperature thresholds. This would provide a much more realistic evaluation of the PSC detections versus altitude and temperature. But then I ask, is the analysis presented in Figure 4 even necessary for this paper? What is the purpose of this analysis?

65) Section 4.2: General comment- why only show one sample PSC event? Now that you have selected a classification scheme- why not process all 14 years of data and produce statistics on interesting aspects of the PSCs such as the mesoscale characteristics? One example is OK- but how representative is it? A statistical analysis would be very interesting and much more compelling for inclusion in the paper. Can you do this?

66) Line 320: Figure A2- I think you are actually referring to Figure A1 here. Why put the model analysis in an appendix? I think it is OK to include in the main text.

67) Line 324: Again think you mean Figure A1

68) Line 325: "fully validate the model" is a strong claim- maybe "The model produces PSC at the 435 K and 475 K levels, and no PSC at the 550 K level above DDU, consistent with the lidar measurements."

69) Line 332: "… temperature history cannot be accounted for." I don't think there is a limitation that temperature history cannot be used and in fact future schemes may indeed include temperature history as a parameter. Therefore, I suggest rewording as "… temperature history has not been accounted for."

70) Line 334: RT and RICE should be written as $R_T$ and $R_{ICE}$

71) Figure 5 discussion: Are the Tice thresholds based on 5 ppmv H2O? This is not likely representative of 28 August when the stratosphere may be severely dehydrated. How would this change your interpretation?

72) Figure 5: hard to see the temperature differentials in the bottom panel- would be helpful to reduce the range of the color bar.

73) Figure 7 discussion: I assume the temperature differences are based on the reanalyses data from over DDU (interpolated) and the temperature at the true location of the radiosonde downwind from DDU- right?

74) Line 379: "NCEP is obviously less accurate …" Is it obvious? Differences with the radiosonde are larger- but doesn't necessarily imply NCEP is wrong. Do you have a citation that NCEP is less accurate than ERA5 and IASI?

75) Lines 388: This statement should appear in the Introduction and help define the focus of the paper!

76) Line 395: "Both NCEP and ERA5 …" You concluded in the previous section that the NCEP temperatures are not as accurate as the ERA5 and therefore the ERA5 data would be used in this study. Why include the NCEP data here?

77) Lines 399-406: How was $T_{NAT}$ calculated for the trend analyses? Did you use a fixed value of 10 ppbv over all altitudes and days? Would a more accurate value reflecting denitrification change your results?

78) Trend analyses: Did you consider subsetting the CALIOP PSC data from 2007-2020 to the DDU location and compare this with the temperature time series? This would be a very interesting exercise to include here.

79) Line 435: "… ERA5 slightly overestimates …" This conclusion is based on comparisons with radiosondes not necessarily collocated with DDU due to balloon drift. Doesn't the drift of the balloon make it difficult to conclude anything quantitative about accuracy since there are spatial temperature gradients that introduce differences?

80) Figure 9: I find the figure confusing- but maybe I just don't understand what is being shown. For PSC thickness between about 2-7 km, the $T<T_{NAT}$ domain is smaller than the PSC thickness? But you state that the figure shows that the $T<T_{NAT}$ domain is significantly larger than the actual PSC thickness! What am I missing?

81) Section 5 Conclusions: No specific issues at this point. Will reserve comment for the anticipated revised version.

---

## Referee Comment (RC3)

The paper by Tencé shows 14 years of PSC observation from Dumont d'Urville. However, the title of the paper is misleading. From the title and introduction, one expected a more detailed presentation and analysis of the observed PSC above Dumont d'Urville. Since PSC observation by ground-based lidars are rare in Antarctica a more in-depth presentation of the dataset would be of interest for the community. The lidar at Dumont d'Urville is in operation since 1989, why do you only focus on the last 14 year? Did the occurrence of PSCs in general and of different PSC types changed since 1989? Do you observe a trend since 1989?

The paper need major revision and the authors need to add a more detailed analysis of the data set.

Major comments:

- It would be good if the authors could add a paragraph about the lidar measurements including some statistics, e.g. How many days per year the lidar was operational? How many PSC were observed per year? When were there observed? Are the PSC observation evenly distributed over the winter?...
- Please include a more detail analysis of the 14-year dataset: e.g. year to year variability, comparison to other ground-based stations and CALIPSO
- Section 4.1:
  - The difference in ICE PSC occurrence needs more discussion. Tesche et al. (2021) shows an ICE occurrence of around 15% compered to 3.7% shown with P18. Could you please provide a pie chart from DDU for the same time period used in Tesche et al. (2021).
  - Also, it would be good if you could provide a plot showing PSC occurrence per year (e.g. histogram). I would assume the ICE occurrence varies from year to year and that it was higher in the years 2015 and 2020, when the ozone hole set record sizes (see Stone et al. 2021).
- Section 4.4. Why do you decide to use NCEP here? In the section before you concluded that NCEP has a T bias from 2k and ER5 should be used. ER5 and IASI should be used here.

Minor comments:

- Line 279: What threshold is used for background aerosol?
- Line 280: Sentence starting with: *This might….. .* Not clear what is being referred to.
- Line 277: If you have a 3% crosstalk at DDU, would that not shift all the observation in Figure 3a slightly to the upper right corner?
- Line 282: *prominent share of NAT class among global* Please add citation.
- Line 285: not sure what is meant by that statement
- Line 284: not clear. B05 has a small occurrence of pure NAT clouds, but NAT is included in MIX. And MIX is quite high in B05.
- Figure 3. Add measurement date in the figure caption. Also, pleas add $d_{aer}$ to B05.
- Line 305. Misleading. MIX is mixture if different types (STS, ICE and/or NAT), not a completely different typ of PSC
- Figure 4.

- o Subplot a and b look the same, as well as d and e. Does that mean that STS and NAT+MIX between B05 and P11 agree very well?
  - o The bimodal distribution for ICE in B05 is very interesting. Could you provide the mean $d_{aer}$ for the peaks?
- Line 345: Really just the drift, not also the different resolution?
- Figure 5a: Looks very smooth. Too much interpolation? And the signal just above the tropopause is not classified. Why?

[Figure]

- Line 384: For the temperature comparison. What resolution were used? Did you interpolate the radiosonde profile on ER5, IASI, and NCEP?
- Section 4.4 and Figure 8. A discussion about the year to year variability of the ozone hole and PSC occurrence should be added here. For example, you could add a second y-axis showing the average ozone hole area for every year.
- Figure 9: What would be the thickness for your definition ($T-T_{nat} < -2k$). Would the model thickness agree than better with the observation?

General comments:

- Figures are to small and of low quality

Stone, K. A., Solomon, S., Kinnison, D. E., & Mills, M. J. (2021). On recent large Antarctic ozone holes and ozone recovery metrics. Geophysical Research Letters, 48, e2021GL095232. https://doi.org/10.1029/2021GL095232

---

## Author Comment (AC1)

We thank the reviewer for their comments. We include here a point by point response in blue.

We draw the reviewer's attention to the fact that, following a similar comment of all reviewers, a figure 1 has been added to the manuscript, featuring information on the operation statistics of DDU lidar. Therefore, all figure numbers have been incremented accordingly as compared to the first version of the manuscript.

Also, to address another comment, the threshold temperatures $T_{NAT}$, $T_{STS}$ and $T_{ICE}$ are now computed based on the closest MLS $H_2O$ and $HNO_3$ concentration measurement. Although this does not change their meaning or their interpretation, most of the figures have gone through some slight changes.

General comment 1: The article is missing a good documentation of the ground-based lidar dataset it is built upon. What is the period of observation covered? How frequent, how long are observation periods? Are specific months (JJAS?) selected, and the rest ignored? Are there annual/seasonal/hourly changes in operation and sampling? How do the sampling coverage and statistics compare to those of CALIOP? This alone could explain differences in ground-based vs spaceborne retrievals.

Following similar comments from other reviewers, the manuscript now includes more detailed information of the operation statistics of DDU lidar. The newly added figure 1 (shown below) presents the number of measurement days per year, from 2007 to 2020, as well as the average monthly distribution of these measurements and their duration.

[Figure]

Figure 1: Operation statistics of DDU lidar. (a) Number of measurement days per year, from 2007 and to 2020 and (b) mean duration of measurement sessions per month, in minutes, from 2007 to 2020.

Apart from exceptional PSC detection in October, PSC are not expected outside the months of JJAS. Concerning the types distribution of figure 3 and the trend of figure 9, only the months of June to September (included) were considered. The first reason is to avoid "false" PSC detection to due to aerosol layers: as Tencé et al. (2022) discussed it, aerosol layers injected in the stratosphere by volcanic eruptions or wildfires can present overlapping optical properties with PSCs. The second reason is that this JJAS restriction is also used by Snels et al. (2021) when comparing groundbased and spaceborne lidar measurements, and we mimic this approach to enable the comparison of our results. The restriction to JJAS for the types distribution of Figure 3 was not explicitly mentioned in the manuscript, it is now added line **312**: "To make the comparison valid, we restricted our analysis to the months of June, July, August and September."

The manuscript originally included a mistake we made interpreting in Tesche et al. (2021). To address this issue and following the reviewers' suggestion, we included in this revised version a

comparison of lidar DDU PSC detections to CALIOP PSC measurements around DDU, from 2007 to 2020.
Please refer to the new figure **3d** and associated analysis. CALIOP is now introduced in section **2.2**, along with its PSC detection method in section **3.2**.

Here is the new version of figure 3:

[Figure]

Figure 3: PSC types distribution observed at DDU for the three considered classifications: B05 (a), P18 (b), P11 (c) and observed by CALIOP extracted around DDU using P18 scheme (d).

General comment 2: The text in most figures is small enough to be sometimes illegible. It would be better if most figures were displayed full-width, but the text would probably remain too small. This is particularly true for figures 1, 2, 3, 4, 7 and 8, in which number axes are extremely small. See the image in the supplement that shows some article text (up left) and a bit of figure 1. Please fix this and make text readable in figures.

General comment about the figure size and quality: this point was also raised by another reviewer. The size of the figures is set by the ACP LaTeX template (12 cm for 2-columns figures, and 8 cm for 1-column figures). They are now larger in the revised manuscript.

**Other comments:**

1) l. 17 (and others): here you refer to PSCs as "stacks of layers". My impression is that it is a very lidar-centric view. PSCs are 3-dimensional structures, as such they can be viewed as stacks of layers, but they could also be viewed as columns of vertical slices, arrays of cubes, etc. I'm not sure what this particular way of describing 3-dimensional structures brings to the table. Please clarify: is there something in the nature of PSC formation and dynamics that leads to a structuration of overlapping, horizontally-consistent slabs? (note this is definitely not true for wave PSCs)

It is a very interesting comment and this layered way of describing PSC clouds is actually lidar-centric ; on the edges of the polar vortex, air masses often exhibit numerous filamentary structures mainly fitting by isentropic layers. Maybe the location of the station warps our description to some extent. For clarity purposes and not to shift the discussion on the geometrical discussion, we chose to remove the phrasing.

Edited line **18**: "... often observed as layers featuring different chemical compositions."

2) l. 46: Later... (Larsen, 2000). Please check the chronology of your paragraph here
The later was referring to the early classifications in 1988 and in the 1990s, but it is indeed not clear with the two previous sentences. To fix the chronology, we removed the "Later," line **55.**

3) l. 63: "different set"
Corrected, edited "different sets" line **72**

 4) l. 65: "we decided to consider 3 different classifications proposed by Blum, Pitts and an updated version of P11 is also considered" -- please fix phrasing: the updated version of P11 either is one of the three different classifications, OR is also considered, but not both.
Corrected, "is also considered" was removed line **76**

5) l. 65: "Following their conclusions": whose conclusions? Achtert and Tesche 2014 are quite far away, please clarify.
It is indeed referring to Achtert and Tesche (2014), the manuscript was edited to make it explicit: "Following the conclusions of Achtert and Tesche (2014)" line **74**

6) l. 68-79: here only sections 1 to 3 are mentioned. Please include all sections. The lidar instrument is actually presented in Section 2, not section 1 as the text says. Processing and schemes are described in section 3 (not 2), etc.
The whole paragraph was rewritten as recommended by another reviewer. It now lists the sections more clearly – and without mistakes, see line **86-94**.

7) l. 83: "(NDACC"
Corrected, parenthesis closed line **98**

8) l. 96 and elsewhere: it looks like you've chosen to use "scattering" where I would have expected "backscattering". Is there a reason for this? Could you clarify in the text that this is your meaning?
It is indeed referring to backscattering, the confusion probably comes from the fact the "Scattering Ratio" is often used to refer to the backscattering ratio. I adapted the manuscript to explicitly refer to backscattering, edited line **113**: "backscattering"

9) l. 96: you defined the backscatter/scattering ratio profiles as the ratio of total scattering to molecular scattering. Is any of those two attenuated? Please be explicit.
None of them are actually attenuated, it is the purpose of what is referred to as lidar data processing or inversion, i.e. getting the backscatter coefficient without. Attenuated backscatter ratio is only used as an approximation to check on the clear sky or an aerosol/cloud signature status of a profile. Actually, the word "attenuated" in this case was a misleading so we replaced it. The preprocessing approximation is to compute a backscatter ratio "not corrected for extinction" (new wording), meaning we consider in the lidar equation the particulate extinction coefficient to be 0. Otherwise, the actual lidar inversion procedures ensures correction of the Mie extinction. The manuscript was edited to remove the two references to "attenuated" and to mention this non-corrected for extinction backscattering ratio line **130-31** and **190**: "... non-corrected for extinction backscattering ratio"

10) l. 101: "data is" plural
Corrected, "data are" line **133**

11) l. 110: "Each instruments"
Correct, plural removed line **151**

12) l. 137: "NCEP reanalysis product is the result of a cooperation between NCEP and NCAR": this info is already provided on lines 127-128.

The repetitive sentence was removed line **182**

13) l. 151: I don't think beta_tot_perp has been defined here yet. I'm guessing that each of the three groups [R_T, R_//] etc is used by a different classification scheme. Please make that explicit.

The three groups are indeed referring to the three classification scheme, the text was edited to make it more explicit: "for each classification scheme" line **197**

beta_tot_perp was indeed not introduced before. However, following the comment of another reviewer, the definition of the variables is now moved to section 2.1, beta_tot_perp is now defined beforehand.

14) l. 165: thanks for the very interesting reference to Behrendt and Nakamura, 2002. I could not find the 0.443% in the text of the article itself, could you expand a bit on how you obtained it? i.e. what temperature or other input parameters you've selected?

The 0,443% parameter is in the table 2 of Behrendt and Nakamura, 2002. We kept the value given for T = 240K, considering with Table 2 and Figure 5 of this paper that the impact of temperature of the molecular depolarization could be neglected for our application.

Since the data is taken directly from the mentioned reference, we do not add any further information in the updated version of our manuscript.

15) l. 169: this was already stated line 151

It was indeed repetitive and the sentence was removed.

16) l. 171: "PSC classification is challenging as described in the introduction but critical": weird phrasing. Please rephrase as e.g., "As described in the introduction, PSC classification is challenging. It is, however, critical..."

Rephrased as suggested by the reviewer "As described in the introduction, PSC classification is challenging. It is, however, critical" line **225**

17) l. 176: "Achtert and Tesche..." the same sentence is already more or less present on page 3

It was indeed stated before in the article, and was therefore rephrased as follows, lines **230-232**: "As mentioned in introduction, the classifications B05 and P11 are considered here following the conclusion of Achtert et al. (2014), to which we had P18, the update of P11 published in 2018."

18) l. 199: MX1, MX2

Corrected, "MIX1 and MIX2" line **250**

19) l. 211: It is unclear to me why you consider P11 in addition to P18. Isn't P18 supposed to supersede the P11 algorithm? Are there reasons why anyone who would like today to study PSCs using CALIOP measurements should go for the P11 algorithm? Version 2 of the CALIPSO PSC product is totally based on P18, so anyone who would like to study PSCs using CALIOP measurements is stuck with P18 anyway (unless she's willing to process the classification herself). Could you clarify what is the point of including P11 in the comparison?

We consider P11 as it is involved in the analysis and conclusions of Achtert et al. (2014), stating B05 and P11 are comparable when applied to the Esrange lidar database with satisfying result. When applying P18 to our groundbased dataset (formerly using P11), we had to adapt the classification so comparing the outcomes of B05, P11 and P18 at DDU sounds consistent to us. It is also a way to validate the adjustment of P18 thresholds to our groundbased setup.

Moreover, as P11 and P18 as well as their intercomparison in Pitts et al. (2018) are based on CALIOP measurements, we consider the additional use of a groundbased dataset interesting.

Finally, the optical properties used in P11 and P18 are different, so it is interesting to us to have

both.

In addition, investigations on PSC with CALIOP data do not necessarily have to be done through P18. CALIOP provides the optical lidar properties with backscattering ratio, backscattering coefficient, aerosol depolarization among other variables (i.e. level 2 products), on which the classifications rely in the end. For the convenience of the scientific community and especially to make an easier link to the model community, the CALIOP scientific team also provides a PSC Mask product relying on P18 and we included in this revised version of the manuscript analysis using this level 3 product.

20) l. 237: "features"
Corrected line **285** "feature"

21) l. 242-244: unclear, what are you planning to do with those mixed-phased clouds? Are you going to make them appear as a separate entity, or subsume them in the category of the dominant particle type, or something else?
In the paragraph mentioned by the reviewer, we try to highlight that the "MIX" clouds defined by classifications correspond to different things i.e. physical/chemical reality. In B05, MIX is defined as any cloud not corresponding to the three other types. In P11, "MIX1" and "MIX2" are defined as different kind of NAT mixtures. Finally, in P18 these two types are merged in a "NAT mixtures" category. It is important that the reader keep in mind that "MIX" is not a tag referring to identical things across the different classifications.

For the purpose of our study, when comparing the PSC types distribution resulting from different classification, we merge the categories referring to NAT clouds and mixed phase clouds to put a common ground for comparison.

22) l. 248: "A distribution of PSC types... published in Tesche et al was included": How did you get the numbers from Tesche et al. 2021? As far as I can tell, the article itself did not include numerical values for its retrievals, so did you lift numbers from the figures? If so, it is surprising you can reach precisions like 15.8%.
The exact numerical values were provided on request by Matthias Tesche directly, this should have been made explicit.

23) l. 270-274: From what you write here I understand that ice PSC are under-represented in DDU lidar PSC observations. If that is indeed what you meant, could you please spell it out explicitly? This actually could be checked (relatively) easily -- in each CALIOP profile one could see for a given PSC type the frequency of opaque tropospheric clouds underneath. According to your explanation, opaque tropospheric clouds should be relatively more frequent in presence of ice PSC than in presence of other PSC types. If you think this is outside the scope of the present paper, perhaps mention it as a possible perspective.
It is outside the scope of this paper but we find it is a very interesting and somewhat necessary perspective. As mentioned in the paper, Adhikari et al. (2010) and Achtert et al. (2012) already explored the correlation of ICE PSC occurrences and tropospheric cloudiness, and it could be interesting to relate this study at DDU especially considering that, as Tesche et al. (2021) highlighted it, DDU experiences a higher level of tropospheric cloud cover hindering its spaceborne validation capabilities.
It is somehow mentioned as perspective in the conclusion, but we decided to strenghten this point .
Edited line **543**:

"... Investigating this correlation is an interesting perspective of this work."

Following the comment of another reviewer, and as CALIOP PSC detection around DDU are now included within this paper, we considered ICE PSC occurrences detected by CALIOP above DDU and crosschecked them against the DDU lidar operation on the same days. In most cases, the groundbased lidar was not operated on these days. The manuscript now includes the following comment, lines **335-338:**

"Between 2007 and 2020, CALIOP detected ICE PSC above DDU on 19 different days, out of which 4 correspond to DDU measurements, suggesting a possible important tropospheric cover or bad weather condition hindering operations. However, we do not consider this small sample robust enough to support the analysis."

24) l. 272: "Marginal" According to your discussion, CALIOP results should be closer to the correct number of ice PSC, and they report a frequency of 16% for ice PSC. Is that marginal?
As stated in a previous comment, there was a mistake in our understanding of Tesche et al. (2021), only based on the winters of 2012 and 2015 as far as Antarctica is concerned. 2015 is a specific year where high ICE occurrences were observed. The updated version of figure 3d features a PSC type distribution based on CALIOP measurements above DDU from 2007 to 2020. It presents an ICE proportion of approximately 10%.
While this 10% share seem not marginal, Pitts et al. (2018) publishing a PSC type spatial distribution where, at DDU location, ICE are not expected to be observed frequently.

25) l. 301: It would be interesting to apply the various classification schemes on the entirety of the CALIOP observations, and indentify in what geographical regions the results diverge. This is clearly outside the scope of the current paper.
The reviewer points out a very interesting study, and we agree that it is not the scope of our paper focused on DDU lidar observations.

26) The discussion of the comparison suggests to me that outputs of classification should come with some kind of reliability indicator, that would decrease as the measured optical parameters get closer to category boundaries. Such an indicator would improve comparisons and make inconsistencies between retrievals perhaps less significant. Is something similar already present in any product? If you think this is a good idea, you could take the opportunity to suggest it in your paper.
That would be an interesting feature for future classifications. Accounting for the uncertainties on PSC class transitions is actually difficult due to complex nature of optical properties modelling from solid particles, but still could be done to some extent. CALIOP metadata actually provide reliability or confidence indices on the PSC mask product.

From the CALIOP website: "These indices provide information on the statistical confidence in the assigned composition based on a data point's location within the optical space. Indices are reported as the distance (in number of standard deviations) between the point and the relevant boundaries of its composition class, with larger numbers indicating higher confidence in the assigned composition."

While these confidence indices are of great use on the massive raw volume of data available through the CALIOP measurements at the continental scale, it remains difficult to set them to practice on the smaller scale of our groundbased dataset of CALIOP overpasses above DDU.

27) l. 327: "This high variability must be kept in mind": why?
The phrasing recalls that classifications tend to present a very global point of view and represent PSC measurements as single points on a plan ($[R_T, R_{//}]$, $[R_\perp, R_{//}]$ or $[R_T, \beta_{tot,\perp}]$ in the schemes

presented here) whereas fine scale measurements as shown in figure 6a highlight the high temporal variability of PSC fields. When using classification outcomes, one should keep in mind the way these values are obtained and the set of parameters controlling their variability (for instance, time integration of lidar measurements, resolution, smoothing, etc).

28) l. 328: "horizontal smoothing... due to the transport" the transport of what? Please clarify.
"Horizontal smoothing due to the transport" implicitly refers to the averaging caused by the integration time and the stratospheric transport during this time window. Maybe rephrasing it makes it clearer, lines **414-415:** "the trade-off on the integration time between SNR and information loss caused by the averaging of potential varying atmospheric scenes due to the air masses transport"
It refers to the fact that, if the atmospheric scene changes during the integration time window (i.e. presence or absence of a cloud for example), the associated integrated optical properties will be representative of a smoothed version of the changes.

29) l. 333: the type changes throughout the whole day, not just once at 5PM. But your point stands.
It is right, I just wanted to point out a specific type change. The manuscript was edited to rather point to the multiple type changes of the PSC layer around 20 km. Line **419**, "around 5PM" was simply removed.

30) l 334: Related to my previous point about a type reliability indicator, do the optical parameters of this cloud hover near the boundary between two categories in the classification diagram? Would an indicator help identify this situation and flag it as unreliable?
We think the subsequent discussion after line **420** provide hints on the reliability of the type change and addresses the point of the reviewer.
Using a reliability index such as the one provided by CALIOP, the type identified for the PSC sample of this figure would have been flagged as less reliable due to the proximity of its optical properties to the boundaries of the NATmix and ICE types in P18.

31) l. 339-341: Could you specify if, in your opinion, these changes in composition (derived from the changes in optical properties) are consistent with the speed of the deposition and growth processes that would drive the change in composition? In other words, are the changes in composition trustworthy, or are they a demonstration of the limitations of the optical classification approach?
The reliability of the composition changes and associated change in the optical properties underlies the question of PSC state at the time it interacts with lidar beam, this being finally related to thermodynamical equilibrium and kinetics of composition changes. As pointed out earlier in the review, this could be considered a lidar centric issue. This comment is really relevant to us : even if this point actually reflects some limitation in the building of classification schemes using optical properties, the only way to circumvent this is to consider the uncertainties as a whole.
Such an advanced classification would need to combine confidence indices such as the ones provided by CALIOP to accurate uncertainty assessment accounting for both lidar signal and air mass thermal history, as PSC formation and composition strongly relies on this parameter.

32) l.354: Here by "lidar" you imply an HSR-capable lidar. Please clarify.
Elastic lidar inversion always requires knowledge of the temperature and pressure to derive molecular scattering. The HSRL technique takes advantage of the spectral distribution of the lidar return signal to discriminate aerosol and molecular signals and thereby measure aerosol extinction and backscatter independently. Our system is not HSRL ready and we derive molecular scattering from external temperature/pressure dataset. Since the DDU lidar is already referenced, we choose not to add any extra information.

33) Figure 5: Here the labels are quite readable, but the decision to make the figure wide and short makes it very hard to identify any structure visually (especially in Figure 5a). Could you please reorganize the figure to change its aspect ratio somehow? Maybe make it a 3-columns/1-row full-width figure?

Addressed in general comment 2 and we are fully aware of this since prepublication, all the figures have been made bigger as the problem was coming from the ACP LaTeX template guidelines. We hope the updated version of the manuscript is more readable now.

34) l. 366-367: "To investigate the effect of temperature variation on PSC..." do you mean "the impact of the choice of temperature dataset on the results of PSC classification"?

We agree on the need to rephrase the relevant lines, a word seems to be missing here making the sentence pretty unclear. Rephrased to "In order to use the most adapted temperature dataset to process our PSC measurements at DDU, we compare several ones in Fig. 8, from reanalysis to satellite observations." lines **443-445**

35) Figure 8: I'm sorry but I don't understand what is being shown here. As I understand it, the figure shows three numbers : A) the number of days in which the lidar observed a PSC (red triangles), B) the number of days in which the ERA5/NCEP temperature allowed PSC formation (green/red crosses), and C) the number of days in which ERA5/NCEP temperatures were 2K below the TNAT formation threshold, AND no lidar measurements were available (grey arrows). In my view, "the number of days in which the ERA5/NCEP temperature allowed PSC formation" is the same as "the number of days in which ERA5/NCEP temperature were 2K below the TNAT formation threshold". In that case, A+C should be equal to B. This is clearly not the case in the figure, so I must have misunderstood something, but I can't find elements in the text to clarify my misunderstanding. Please help.

We note that the stratospheric denitrification is now taken into account in the $T_{NAT}$ computation, and it tends to decrease $T_{NAT}$ values. As a result, the $\Delta T$ criteria adjusted to our lidar measurements is now -1 K and not -2 K as it was the case in the initial version of the manuscript. Figure 9 and the associated discussion have been edited on lines **490** and **492** and in the caption of Figure 9.

The new version of Figure 9 is shown below:

[Figure]

Figure 9: PSC days per year at DDU from 2007 to 2020 featuring PSC detection with the lidar in red triangles. Potential PSC days per year estimated by ERA5, NCEP and IASI based on the lidar measurements are shown in green and red respectively. Green, blue and fuchsia lines represent the corresponding trends. Grey arrows indicate the number of days per year where the T - $T_{NAT}$ < -1 K criterion was satisfied and DDU lidar was not operated.

Figure 9 is indeed showing these three parameters: number of days with a lidar PSC detection at DDU (red triangles), number of days where T-$T_{NAT}$ < -1K is reached with ERA5 and NCEP (green and blue crosses), and the number of days where this criterion is reached with ERA5 and NCEP but no lidar measurements was available.

We provide more information on the methodology used to build this trend as follows:This temperature proxy for PSC is adopted to keep the use of the temperature threshold Tnat to predict

PSC formation. As mentionned in the litterature (Dye et al., 1992 for example), PSCs usually form a few degrees below $T_{NAT}$, so that using $T-T_{NAT} < 0K$ as a criterion leads to an expected PSC overestimation.

The groundbased dataset provides the number of days where the lidar was operated as well as the number of days where PSC were detected at DDU.

Independently, we calculated the number of days where the threshold $T-T_{NAT} < Delta$ was reached on the lidar operating days, spanning the Delta range -10K to 0K. We then used this delta as criterion to match the number of PSC days detected by the lidar to the number of predicted PSC days. In other words, the delta variable is used as control variable between two criteria: the one of our lidar observations and the one, using $T_{NAT}$, of the model renalyses. But the trend is in essence built from PSC detection using lidar measurements.

To answer the reviewer's question, the sum of red triangles and grey arrows does not necessarily add up to the green / blue crosses count. First, the method is designed so that, when only computed for the dates where the lidar was operated, the green / blue crosses are as close as possible to the red triangles. But the distance between both datasets (crosses and triangles) was of course optimized for the 14 years, not individually for each year. Crosses and triangles would have been exactly equals should we have computed a specific Delta value for each year, which would render the trend meaningless.
From a more qualitative point of view, it is also expected that triangles + arrows do not equal the cross values. Sometimes, $T-T_{NAT} < -1K$ is reached but no PSC is formed: the temperature is not a sufficient condition for PSC formation. On the other hand, some days we detect a PSC but ERA5 or NCEP state that $T-T_{NAT}>-1K$: this is for example the case if a PSC is formed following a subscale cooling temperature not resolved within the models.

Finally, please keep in mind that this method, i.e. the adjusement of a criterion to derive a number of PSC days per year, is in essence a statistical approach and not a theoretical formation criterion calculation. It is expected that the year to year variability makes the criterion over or underestimated because it is set as the best match considering the 14-year dataset as a whole.

Please note that this figure is edited according to other reviewers' comments. It now includes the same trend computation based on IASI temperature measurements (marked as fuchsia crosses and line), as well as the number of PSC days detected by CALIOP above DDU as black triangles.

36) l. 408: the negative trend that is found here mostly depends on the reliability of ERA5 and NCEP stratospheric temperatures, and on the presence of a overall stratospheric temperature trend in those datasets, correct? Could you make it clearer why your results are not just confirming the presence of a warming trend in ERA5/NCEP stratospheric temperatures? i.e. what is the lidar bringing here?
In light of the details on the previous comments, we hope the role and added value of the lidar measurements in the trend is made clear by now. To address the second point, the trend shown here is of course connected to temperature trends, but not equivalent. There is a difference between a trend of mean temperatures, and a trend of overpassing a given threshold. Figure 9 is the latter one. We could have a stratospheric warming trend that does not necessarily impact the extreme values, and vice versa. As for PSC formation, the occurrences of extreme temperatures under a given threshold remain the critical factor, highlighted in Figure 9. So we consider that the trend of Figure 9 is not just confirming a warming trend, it is connected to it but it brings a somewhat different

information.

37) l. 445-448:I understand from your conclusions that 1) applying the three classifications schemes to ground-based lidar observations leads to results that agree quite well, and 2) applying the same classification scheme (P18) to ground-based and spaceborne lidar leads to results that agree well too. From this, I understand that the choice of classification scheme has after all little importance on the results. Do you share that opinion? If not, could you amend your conclusions to include arguments for the opposite viewpoint?

The overall agreement on using different classifications considering one of them is built from an arctic dataset (B05) and the other from a spaceborne dataset (P11, P18), i.e. built on a different scale and relying on different optical variables, should be seen as a successful characterization of complex optical patterns using different observational geometries, carrying different uncertainties. We somewhat share the opinion stated in the reviewer's comment, but it is worth considering the scale of the dataset on which the classification is applied. On this decadal scale, an overall agreement is found. For process studies, and especially considering aerosol plumes, particle characterization may be more tricky as optical properties can be closer to the boundaries of any given scheme, leading to, on the smaller scale, inaccurate characterization.

The choice of the classification may also depend on the instrumental setup. The different classifications rely on different variables and some of these variables are directly accessible while some have to be undirectly computed, this leading to larger uncertainties. We aim at providing the community with a DDU dataset carrying its own features and highlights, with a dedicated set of optical properties. Considering our low number of ICE PSC detection, the three classifications overall show a very good mutual agreement.

**References (added references to the manuscript are listed in bold):**
Achtert, P., Karlsson Andersson, M., Khosrawi, F., and Gumbel, J.: On the linkage between tropospheric and Polar Stratospheric clouds in the Arctic as observed by space–borne lidar, Atmospheric Chemistry and Physics, 12, 3791-3798, https://doi.org/10.5194/acp-12-3791-2012, 2012.

Adhikari, L., Wang, Z., and Liu, D.: Microphysical properties of Antarctic polar stratospheric clouds and their dependence on tropospheric cloud systems, Journal of Geophysical Research: Atmospheres, 115, https://doi.org/https://doi.org/10.1029/2009JD012125, 2010.

**Ansmann, A., Ohneiser, K., Chudnovsky, A., Knopf, D. A., Eloranta, E. W., Villanueva, D., Seifert, P., Radenz, M., Barja, B., Zamorano, F., Jimenez, C., Engelmann, R., Baars, H., Griesche, H., Hofer, J., Althausen, D., and Wandinger, U.: Ozone depletion in the Arctic and Antarctic stratosphere induced by wildfire smoke, Atmospheric Chemistry and Physics, 22, 11 701–11 726, https://doi.org/10.5194/acp-22-11701-2022, 2022.**

Dye, J. E., Baumgardner, D., Gandrud, B. W., Kawa, S. R., Kelly, K. K., Loewenstein, M., Ferry, G. V., Chan, K. R., and Gary, B. L.: Particle size distributions in Arctic polar stratospheric clouds, growth and freezing of sulfuric acid droplets, and implications for cloud formation, Journal of Geophysical Research: Atmospheres, 97, 8015–8034, https://doi.org/https://doi.org/10.1029/91JD02740, 1992.

**Rieger, L. A., Randel, W. J., Bourassa, A. E., and Solomon, S.: Stratospheric Temperature and Ozone Anomalies Associated With the 2020 Australian New Year Fires, Geophysical Research Letters, 48, e2021GL095 898, https://doi.org/https://doi.org/10.1029/2021GL095898, e2021GL095898 2021GL095898, 2021.**

Snels, M., Colao, F., Cairo, F., Shuli, I., Scoccione, A., De Muro, M., Pitts, M., Poole, L., and Di Liberto, L.: Quasi-coincident observations of polar stratospheric clouds by ground-based lidar and CALIOP at Concordia (Dome C, Antarctica) from 2014 to 2018, Atmospheric Chemistry and Physics, 21, 2165–2178, https://doi.org/10.5194/acp-21-2165-2021, 2021.

Stone, K. A., Solomon, S., Kinnison, D. E., and Mills, M. J.: On Recent Large Antarctic Ozone Holes and Ozone Recovery Metrics, Geophysical Research Letters, 48, e2021GL095 232, https://doi.org/https://doi.org/10.1029/2021GL095232, e2021GL095232 2021GL095232, 2021.

Tencé, F., Jumelet, J., Bekki, S., Khaykin, S., Sarkissian, A., and Keckhut, P.: Australian Black Summer Smoke Observed by Lidar at the French Antarctic Station Dumont d'Urville, Journal of Geophysical Research: Atmospheres, 127, e2021JD035 349, https://doi.org/https://doi.org/10.1029/2021JD035349, e2021JD035349 2021JD035349, 2022.

Tesche, M., Achtert, P., and Pitts, M. C.: On the best locations for ground-based polar stratospheric cloud (PSC) observations, Atmospheric Chemistry and Physics, 21, 505–516, https://doi.org/10.5194/acp-21-505-2021, 2021.

---

## Author Comment (AC2)

We thank the reviewer for their comments. Please find below our point by point response in blue.

We draw the reviewer's attention to the fact that, following a similar comment of all reviewers, a figure 1 has been added to the manuscript, featuring information on the operation statistics of DDU lidar. Therefore, all figure numbers have been incremented accordingly as compared to the first version of the manuscrit.
Also, to address another comment, the threshold temperatures $T_{NAT}$, $T_{STS}$ and $T_{ICE}$ are now computed based on the closest MLS $H_2O$ and $HNO_3$ concentration measurement. Although this does not change their meaning or their interpretation, most of the figures have gone through some slight changes.

Concerning the comparison of our PSC record with other Antarctic groundbased datasets, it is right that it should have been included and it is now the case. DDU PSC distribution is compared with PSC types distributions produced at McMurdo and Concordia by Snels et al., (2019, 2021), a new paragraph has been added in the manuscript, lines **341-354**:
"At McMurdo antarctic station (77.85°S - 166.66°E), from 2006 to 2010, Snels et al. (2019) reported a mean distribution of 13.8% STS, 71.6% NATmix, 2.6% ENH and 12% ICE. During this period, CALIOP observed approximately 10% more STS and 10% less NATmix and otherwise shows a good overall agreement with the groundbased lidar. Snels et al. (2021) compare Concordia groundbased PSC detections to CALIOP measurements around the station from 2014 to 2018. This study mainly focuses on the agreement between groundbased and spaceborne instruments and do not directly provides the PSC types distribution observed at Concordia by the local lidar, but still shows the distribution observed by CALIOP around the station. From 2014 to 2018, the yearly occurrences of STS represent from 14 to 38%, NATmix from 42 to 67%, ENH from 5 to 11% and ICE from 10 to 28%. The distribution shows a high annual variability, but we still can point out differences with DDU PSC types distribution. Both McMurdo and Concordia measurements feature a higher proportion of ICE detections and less STS observations as compared to DDU. The 38% STS share observed at Concordia in 2014 is considered to be an outlier. This is consistent with the results of Fig. 19 of Pitts et al. (2018) which show that main area of ICE occurrence is located inside the continent while DDU is located on the Antarctic coast. The temperature necessary to form ICE crystals are reached less often at DDU, on the edge of the vortex."

Following several comments, a comparison with CALIOP PSC measurements around DDU is now included in the study and is extensively described in this document. It notably relies on the method used by Snels et al. (2021) for Concordia and this study is therefore cited in this context.

1) The current Introduction in Section 1 needs major reworking. What are the main goals of this study? How are the ancillary datasets used? These need to be clearly articulated here. This will provide a roadmap for the rest of the paper. From the current introduction, it's not clear why so much effort is going into evaluating the classification schemes. Is this a major goal of the paper? The last paragraph of the current Introduction is attempting to describe the remaining sections of the paper- but it is not accurate or complete. Please rewrite to better summarize what is in each subsequent section (e.g., 2.-Methods, 3- Lidar data processing, 4- Results, 5-Conclusions).
The introduction has been edited to expose more clearly the goals of our study, lines **80-85**:
"Considering the latest laser source replacement in 2005 and the continuous monitoring from 2006, DDU PSC dataset is compared to the spaceborne PSC measurements conducted by CALIOP (Cloud-Aerosol Lidar with Orthogonal Polarization) in the vicinity of the station, from 2007 to 2020. Two of the major roles of a ground station are to perform process studies and establish decadal trends. Such trends are highly valuable because they reflect the evolution of the stratosphere, in terms of temperature and chemical compositions. In this study, DDU lidar measurements are exploited to produce a trend of PSC days per year at DDU."

The last paragraph has been rephrased to feature more clearly the description of the remaining sections. Edited, lines **85-93**: "Section 2 presents the data and instruments exploited in this study. Section 3 introduces the PSC detection methods as well as the considered classification schemes. Section 4 exposes the results of the study. First, the outcomes of the application of the classification schemes B05, P11 and P18 to the DDU lidar data record are presented and discussed, and CALIOP and DDU PSC measurements are compared. Then the analysis of an interesting example of a long lidar session is used to illustrate the unique capabilities of lidar measurements in characterizing very fine vertical features in PSC fields. Third, lidar measurements and temperatures from ERA5, NCEP and Infrared Atmospheric Sounding Interferometers (IASI) are combined to produce a PSC occurrence trend from 2007 to 2020. To support the use of these temperature data sources, they are compared to temperatures from radiosondes launched daily at DDU. Finally, some challenges of PSC parametrization in climate models are discussed before exposing the main conclusions."

2) Line 3: The meaning of the term "tight model parameterization" is not clear. Do you mean mathematically simple and/or computationally fast?
"Tight model parametrization" is meant here in the sense that models need to keep the number of variables used for PSC parametrization as low as low possible.
Tight actually refers to a low number of model variables for parametrization of PSC microphysics. It is following the reviewer's guess related to the degree of mathematical/physical complexity. For clarity edited in the manuscript, lines **3-4** : "...and model parametrization constraints".

3) Line 17: What is meant by "stacks of layers featuring different mixtures." Please cite a reference and describe in a little more detail what is meant by this phrase.

We mean that PSC fields often consist of several layers corresponding to different types of clouds, i.e. different mixtures, thus, different chemical composition. Considering comments from below, layer is defined as consecutive vertical bins of the same type. Illustration is clear on Figure 7 in our paper, Figure 4 in Blum et al.[2005] or Figures 4 to 8 in Snels et al. [2021].
Since this point was raised by several reviewers and was not central to our analysis, it was rephrased as follows, line **18**: "often observed as layers featuring different chemical compositions"

4)Line 18: change "when temperature" to "when the temperature"
Edited: "when the temperature" line **19**

5) Lines 20-22: Strictly speaking, I believe denitrification and dehydration refer to the redistribution and irreversible removal of $HNO_3$ and $H_2O$ from the stratosphere. Uptake of $HNO_3$ and $H_2O$ by itself (through particle formation) may be reversible. Therefore, denitrification and dehydration occur by sedimentation of large NAT or ice PSC particles that contain $HNO_3$ an/or $H_2O$.
That is what we meant but the sentence was poorly phrased, edited the manuscript to be clearer. What was meant is that HNO3 / H2O are taken from the gas phase into PSC particles, which are then removed from the stratosphere as the PSC sediment.
Edited line **22**: "Denitrification and dehydration, mostly through the uptake of HNO3 and H2O by PSCs and the subsequent PSC sedimentation, decrease HNO3 and H2O stratospheric concentration and hence enhance ozone depletion."

6) Line 23: The phrase "a lot" is not a good choice for a technical paper. Would be good to have some citations here on what significant improvements have taken place and what remains to be understood. Perhaps more relevant here, is this study going to improve our understanding of any of these outstanding questions?

The paragraph was edited to include examples of the improvement of our global understanding of PSCs in recent years as well as examples of the PSC mechanisms still actively studied. Edited lines **23**-**32**:

"Despite major improvements in the recent years due to enhanced research and monitoring capabilities, some PSC-related aspects are still to be understood. Spatial measurements brought a global point of view able to grasp the spatial and temporal distribution of PSC during winter (Pitts et al., 2018). Studies highlighted the need to take wave-induced temperature variations in account to adequately model PSC occurrences (Cairo et al., 2004; Höpfner et al., 2006; Eckermann et al., 2006; Engel et al., 2013; Tritscher et al., 2019). However, some PSC particles formation pathways are still debated (Tritscher et al., 2021) and the adequate model parametrization of PSC is still challenging. Besides, the recent stratospheric injections of aerosols caused by volcanic eruptions and wildfires also raise questions on the potential interaction with PSC formation processes and subsequent stratospheric ozone depletion (Tencé et al., 2022; Ansmann et al., 2022; Rieger et al., 2021 Stone et al., 2021)"

7) Line 27: "sulfur aerosols" should be "sulfuric acid aerosols"
Edited: "sulfuric acid aerosols" line **35**

8) Line 27: meteoritic material- is there a citation you could include here that shows meteoritic material may be efficient PSC nuclei?
This is still a subject of discussion, so we moderated our assumption. Section 3.3 in Tritscher et al. (2021) discusses the possibility of meteoritic material to act as nuclei for PSC particles and the mutliple laboratory and fields works published on that topic. We can for example cite the work of James et al. (2018) who proved that meteoritic material can trigger NAT nucleation in PSCs, or the field measurements of Ebert et al. (2016) which suggest meteoritic material could be involved in a PSC nucleation pathway.
Edited lines **35**-**36**: "or to a lesser extent meteoritic material, which role in PSC particles nucleation is still a subject of discussion (Ebert et al., 2016; James et al., 2018)".

9) Lines 27-29: I suggest listing relevant citations in the same order as the particle compositions (ice, NAT, STS) ... (Peter and Grooß, 2012; Hanson and Mauersberger, 1988; Carslaw et al., 1997). Did Peter and Grooß (2012) actually perform lab studies on ice? I thought this was a chapter in a book.
Peter and Grooß (2012) is indeed a book chapter notably reviewing PSC particles nucleation pathways. Koop et al. (2000) may probably be more suited.
Edited line **38**: "'(Koop et al., 2000; Hanson and Mauersberger, 1988; Carslaw et al., 1997)"

10) Lines 30-31: "NAT particles only nucleate on pre-existing particles." - what pre-existing particles? ice? meteoritic material? Citations?
Original intent was to highlight that NAT only nucleates heterogeneously because the homogeneous nucleation is kinetically suppressed (Koop et al., 1995; Koop, Carslaw et al., 1997), and not to discuss the potential nuclei here. NAT nucleation on pre-existing ice particles is established (for example, Koop, Carslaw et al., 1997) but does not explain all NAT observations (Drdla et al., 2002; Pitts et al., 2011).
Citations added line **40**: (Koop et al., 1995; Koop, Carslaw et al., 1997, James et al., 2018)

11) Line 37: The Wegner et al. (2012) Figure 1 only shows efficiencies for liquid aerosol (binary and ternary) and NAT, not ice. Aren't these efficiencies primarily based on the available surface area? Is it really composition dependent or mostly surface area density dependent?

Ice particles are indeed missing from this figure. I mistook it with the adaptation made by Tritscher et al. (2021) that includes ice. I changed the citation.
Edited line **46**: "in Fig. 39 of Tritscher et al. (2021)".

Efficiencies are indeed surface area dependent, the point of Fig. 39 of Tritscher et al. (2021) and Fig. 1 Wegner et al. (2012) is to link particle types, and therefore composition, to chlorine activation efficiencies. I edited the previous sentence to mention the link with surface area density.

Edited line **42**: "Depending on their dominant type of particles, different PSCs have different surface area densities and chemical heterogeneous reactivities..."

12) Lines 38-39: This sentence is confusing to me. What do you mean by "pure STS, NAT, and ICE blends of chemical compounds?" Are you simply referring to the chemical makeup of the particles?
Original sentence is indeed misleading. Intent was to say that PSC classifications were based on the optical properties of pure STS, NAT and ICE along with mixtures of the latter species. Edited lines **47-49**: "From these three basic particle types and their combinations, more detailed types of PSCs were identified for the purpose of creating classifications based on optical properties of the pure STS, NAT and ICE blends of chemical compounds."

13) Line 40: "Poole and McCormick (1988) in 1988." I think it is obvious that Poole and McCormick was published in 1988, so you don't need the additional "in 1988"
This is of course a typo in the bibliography management, it has been addressed throughout the revised manuscript.

14) Line 41: "set" should be plural "sets"
Edited "sets" lines **50**

15) Line 43: "Achtert and Tesche (2014), 2018)." I think this is a typo- ", 2018)" should be deleted.
It is indeed a typo, it was deleted.
Edited: ", 2018)" was deleted line **53**

16) Line 45-46: "... but whose presence was not proven in atmospheric observations."
Suggest rephrasing as "but has yet to be confirmed by atmospheric observations."
Rephrased as suggested.
Edited: "but have yet to be confirmed by atmospheric observations." Lines **56**

17) Lines 46-48: What studies have shown these "stacks of fine layers?" I don't think that the Larsen paper shows that PSCs often occur as stacks of layers of different particle types- at least I didn't see any mention of this in the report. Larsen does conclude that the temperature history of the air mass must be known to properly simulate the particle formation.
The Larsen paper is cited as support of the statement on the history of the air masses.
See comment 3) concerning the stacks of fine layers statement. Since the mention of the stack of layers was not useful here, it is removed to avoid confusion.
Edited line **56**: "Studies showed that PSC type identification depends on the history of the air masses due to hysteresis effects in PSC formation along the temperature scale (Larsen, 2000)."

18) Lines 48-49: Sentence is poorly worded. Suggest something like "In addition, the temperature cooling rate is an important variable driving orographic PSC formation in both the Arctic and Antarctic (Noel and Pitts, 2012)."

Rephrased as suggested, lines **58-59**.
"In addition, the temperature cooling rate is an important variable driving orographic PSC formation in both the Arctic and Antarctic (Noel and Pitts, 2012)"

19) Line 51: "only few" should be "only a few"
Corrected. Edited "only a few" line **61**

20) Lines 51-54: This sentence is not clear and too long. What is based on "optical properties"? The complex observational patterns? Surely not the parameterization schemes? Numerous phrases that are not clear: "tight as possible"? "observations derived patterns"? Please try to reword.
Sentence actually needs rephrasing. Edited at lines **61-63** into: "Therefore, the combination of model constraints and high rate of unclassified observations (due to either instrumental concerns or unequilibrated particles) led to some redefinition of the boundaries between the existing PSC classes rather than considering additional classes."

21) Lines 54-56: This sentence seems repetitive with the sentence L.40-42. Maybe you can combine this with the sentence on L. 40-42 and list the citations there?
The citations were moved in the previous paragraph, lines **50-51**, and the sentence lines **63-64** was rephrased. The idea was to mention that these classifications are based on lidar measurements, whereas the previous paragraph focused on the evolution of PSC classifications following the evolution of our understanding of these clouds.

Edited: "The PSC classifications schemes previously mentioned are based on ground-based or space-borne lidar measurements" Lines **63-64**

22) Line 63: "different set of" should be "different sets of"
Corrected. Edited "different sets of" line **72**

23) Lines 65-67: Years inside the parentheses are not necessary. Suggest rewording "...
Blum et al. (2005) (hereafter called B05), Pitts et al. (2011) (hereafter called P11), and
Pitts et al. (2018) (an updated version of P11, hereafter called P18)."
Typo with the bibliography corrected. The years in the parentheses line **76** were removed.

24) Consideration of P11: The P18 algorithm corrected several know deficiencies in the P11 algorithm. I believe the P18 has replaced P11 as the operational algorithm used to produce the CALIOP v2 PSC data products. Therefore, there is no reason to include the P11 version in your evaluation unless you just want to compare the differences between P11 and P18 (that was done by Pitts et al., 2018). Is that your goal here?
We consider P11 as it is involved in the analysis and conclusions of Achtert et al. (2014), stating B05 and P11 are comparable when applied to the Esrange lidar database with satisfying result. When applying P18 to our groundbased dataset (formerly using P11), we had to adapt the classification so comparing the outcomes of B05, P11 and P18 at DDU sounds consistent to us. It is also a way to validate the adjustment of P18 thresholds to our groundbased setup.
Moreover, as P11 and P18 as well as their intercomparison in Pitts et al. (2018) are based on CALIOP measurements, we consider the additional use of a groundbased dataset interesting. Finally, the optical properties used in P11 and P18 are different, so it is interesting to us to have both.

25) Line 70: suggest changing "station hosts" to "station has hosted"
Edited line **79** "has hosted"

26) Summary paragraph beginning on Line 68: As mentioned above, this last paragraph in the Introduction needs completely rewritten. This paragraph is attempting to describe the following sections of the paper- but is not accurate or complete. Please rewrite to better summarize what is in each subsequent section.

The paragraph is rephrased to better introduce the sections content. Please see comment 1 with the added manuscript discussion in lines **86-94**.

27) Line 80: Section 2 Methods: This section really doesn't describe methods- rather just the datasets used in the study. Probably should rename "Datasets"?

It is true, the section was renamed "Data and instruments", line **95.**

28) Line 82: Suggest changing "Since April 1989, an aerosol/cloud lidar system is in operation at DDU ..." to "An aerosol/cloud lidar system has been in operation at DDU since April 1989 ..."

Rephrased as suggested, line **97**: "An aerosol/cloud lidar system has been in operation at DDU since April 1989 ..."

29) Line 83: Add closing parenthesis after NDACC

Edited ")" Line **98**

30) Line 84: Delete extra space after "Antarctic atmosphere"

Extra space deleted

31) Lines 85-87: Awkward grammar- suggest rewording this sentence to "Although the measurement calendar focuses on the PSC season with nighttime setup, the recent focus on aerosol plumes either originating from volcanic or biomass burning activity (Tencé et al., 2022) has extended the measurement calendar to the summertime."

Rephrased as suggested, lines **100-102**: "Although the measurement calendar focuses on the PSC season with nighttime setup, recent work on aerosol plumes either originating from volcanic or biomass burning activity (Tencé et al., 2022) suggests extending the measurement calendar to the summertime."

32) Section 2.1 DDU Lidar description: You say that the lidar capabilities have been continuously upgraded and cite the David et al. (2012) paper. Have there been any notable upgrades in the past 10 years since the David et al. paper?

Notable technical upgrades concern the redesign of the eclectronical synchronization circuit, software acquisition cards, photomultiplier tubes and emission optical bench. Noticeably, the laser source has been renewed last 2022 summer campaign, and redesign of the channel box is under study.

33) Section 2.1 DDU Lidar description: Since you are introducing most if not all of the lidar optical parameters here, I suggest you move the equations defining the lidar parameters in Section 3.1 to this section. It seems more appropriate to have the definitions here. Maybe after L. 94?

It is indeed more adapted. Variables definitions were moved from section 3.1 to section 2.1. Consequently, two sentences were displaced and one was added lines **114-115** before the variables definitions: "The different lidar related variables used in this study are defined as follows:".

34) Line 97: "saturation effects" - Would you please describe what the saturation effects are and add more detail on how they are removed?

The mention of saturation effects misses context elements: basically, saturation only concerns a very small fraction of the 2007-2020 individual profiles and is mostly related to the new setup with

the laser replacement last year. Saturation leads to a bias in the overall shape of the lidar profile due to photons being missed at the maximum power altitude. Correcting saturation effect is only done with a clear tropopause: indeed, thick cirrus layers will prevent us from rebuilding the molecular shape of the lidar signal. Saturation correction parameters are set from expected Rayleigh scattering calculations and maximum saturation value of the photomultiplier tube.

35) Line 98: "homogeneity of the scene" –How is the homogeneity quantified and used?
The statement was too vague because these pre-processing steps are described in section 3.1, we provided more details. Please see comment 49) which extensively addresses this topic.

36) Line 100: "altitude" should be plural "altitudes"
Corrected. Edited "altitudes" line **132**

37) Section 2.1.1 IASI temperature product: What is the main role of the IASI temperature product in this study? Should note that in the introduction (section 1).
The temperature retrieval from the IASI instrument is recent and provides a sample of daily temperature profiles extracted above the station. This feature is critical to us because it may be more suited to groundbased validation than other spaceborne datasets we could have used with lower relevancy in terms of distance, and measurement frequency. Besides, considering the input of another reviewer, IASI temperature dataset was included in the PSC trend presented in Figure 9. IASI is expected in this study to fill the observational role as temperature provider whereas NCEP and ERA5 are reanalyses. For example, IASI can detect small scale temperature variations relevant to PSC formation that reanalyses product may miss.

38) Line 111: "instruments" should be singular "instrument"
Corrected. Edited "instrument" line **151**

39) Line 112: à should be "a"
Corrected. Edited "a" line **152**

40) Line 113: delete period after PM
Corrected. Period deleted line **153**

41) Line 118: "temperatures" should be "temperature"
Corrected. Edited "temperature" line **158**

42) Line 120: "very good agreement" – Please be more quantitative- what is very good agreement?
Precisions have been added, from Bouillon et al. (2022). Edited, lines **161-164**: "Bouillon et al. (2022) showed that daily zonal mean differences between IASI ANN and ERA5 at mid and high latitudes are lower than 0.5 K between 750 and 7 hPa and reach 2 K at 2 hPa. Comparing IASI ANN with a global radiosoundings archive (Analyzed RadioSoundings Archive), Bouillon et al. (2022) found no significant bias and a standard deviation between 1 and 2 K for the Antarctic region."

43) Lines 124-125: Sentence is worded awkwardly. Suggest rewording something like "As discussed in further detail in the following sections, reanalysis temperature products areoften utilized to complement or replace local radiosonde measurements for both data processing and interpretation of ground-based lidar measurements."
Rephrased as suggested, lines **167-169**: "As discussed in further detail in the following sections, reanalysis temperature products are often used to complement or replace local radiosonde measurements for both data processing and interpretation of ground-based lidar measurements."

44) Lines 129-130: What do these acronyms (4D-Var, Cy41r2) mean?
Cy41r2 is actually the usually refered to name of the Integrated Forecasting System (IFS) model cycle used by ECMWF.
4D-Var stands for "four-dimensional variational data assimilation". It is now specified in the manuscript. Edited line **173** "four-dimensional variational data assimilation"

45) Line 133: "interpolated at DDU location" – "interpolated to the DDU location" Is it simply linearly interpolated from the original product grid (0.25 x 0.25 degree)?
It is indeed linearly interpolated from the original grid.
Edited line **177**: "to the"

46) Line 135: "dynamic tropopause"- how do you define the dynamic tropopause? Is this an ERA5 product that you interpolate to the DDU location?
The tropopause is defined as the lowest point between the 380 K potential temperature and |PV| = 2PVU. As the potential vorticity is not available in ERA5, it is calculated from the vorticity, the winds and the temperature.
This is now made clear as follows, lines **180-181**: "dynamic tropopause, defined as the lowest point between the 380 K potential temperature and |PV| = 2PVU (calculated from the vorticity, the winds and the temperature from ERA5)"

47) Line 139: NCEP product: What is meant by "provide an output for DDU"?
The NDACC/NOAA archive mentioned in the data availability section used for NCEP data provides daily profiles of temperatures and geopotential heights on 18 pressure levels from 1000 to 0.4 hPa at NDACC stations, which include DDU. Rephrased line **185** for better clarity into "provides direct extraction of the temperature product above DDU"
https://www-air.larc.nasa.gov/pub/NDACC/PUBLIC/meta/ncep/ncep_2022.pdf

48) Reanlyses Data discussion in general: Again, there should be some discussion in the Introduction of how the reanalyses data will be used in this study. What are the uncertainties in reanalyses data products?
NCEP provides an error estimation of the temperatures at DDU of 2.5 K from the surface to 50 hPa, then 3.5 K at 30 hPa and 10 hPa and more than 5 K above 5 hPa. The errors on the geopotential heights provided at DDU range between 16 m near the surface to 150 m at 5hPa.
ERA5 uncertainties are discussed at:
https://confluence.ecmwf.int/display/CKB/ERA5%3A+data+documentation#ERA5:datadocumentation-Accuracyanduncertainty but no specific uncertainty field is provided for in the ERA5 product.

For Antarctica, ERA5 validation in the Southern Antarctic suggests a warm bias of 0.14 °C, with a significant improvement compared to the ERA5 predecessor, ERA-Interim, in particular at high altitudes (Tetzner, D.; Thomas, E.; Allen, C. A Validation of ERA5 Reanalysis Data in the Southern Antarctic Peninsula—Ellsworth Land Region, and Its Implications for Ice Core Studies. *Geosciences* 2019, *9*, 289, doi:10.3390/geosciences9070289).

Our study does not aim at validation nor assessing the quality of the reanalysis. Moreover, since it is very complicated to discuss these biases and how they evolve temporally/seasonally, and in particular over the location of DDU, we choose not to discuss the reanalysis uncertainties in the new version of the manuscript. More information on the intercomparison between NCEP and ERA5 is heavily listed in comments 74 and 79 hereafter.

49) Section 3.1 PSC detection by lidar: I found the description of the PSC detection here to

be confusing. The first step is some "pre-processing" that identifies time segments that contain aerosol/cloud? What is dynamic time averaging? What do you mean by "next step summation according to homogeneity"? What is the "peak detection algorithm"? Are you just searching each profile identified as containing aerosol/cloud for peaks that identify layers? The output of the detection algorithm are profiles of lidar parameters with one or more layers identified as being aerosol/cloud? You mention in Lines 150-152 that a type is attributed to each layer. Isn't the composition classification performed separately from detection and dependent on the specific scheme being used as described below in Section 3.2? Please try to rewrite this section more clearly with more detail. This paragraph has been reworked, we hope the reviewer will find it clearer. "dynamic" here was a poor wording and was removed. The idea is that the time averaging does not split the measurement sessions blindly, but takes the atmospheric situations sounded into account:

The purpose of the pre-processing step is to avoid blending in different atmospheric scenes when choosing the integration time windows. Averaging 30 min of measurements which contain 15 min of measurements of a cloud and 15 min of clear air will result in a somewhat smoothed signature that does not clearly point to any physical observational reality. To avoid this effect, we first perform a first-order inversion assuming no particulate extinction on every 15 minutes file. The resulting approximated scattering ratio profiles are then flagged to state on aerosol / cloud presence. Doing so, if for instance a measurement session captures an atmospheric change like the appearance / disappearance of a cloud, we are able to cut the session at the time of this change to avoid blending in different situations. This is what we call a homogeneous atmospheric scene: a time window during which the stratospheric column sounded by the lidar is globally stable signal-wise.

After this step, the raw 3 minutes files are integrated following the results of the pre-processing algorithm and proper signal inversion is performed. Once the $R_T$ and $\delta_{aer}$ computed, a peak detection algorithm is applied, identifying the peaks in the signal. Peaks identification is a typical feature in signal processing with many available implementations, the challenge being to separate outliers using input parameters not specific to our instrument. Peaks detected in the $R_T$ and $\delta_{aer}$ profiles correspond to scattering layers. Once the layers have been identified, the relevant parameters are computed for each layer. As mentioned in the manuscript, there are three sets of such parameters, which correspond to the three classifications considered in this study. Therefore, the result of this detection algorithm consists of three sets of layers and their optical parameters, corresponding to the three classifications B05, P11 and P18. The altitude and [$R_T$, $\delta_{aer}$] of the layers are the same in each of these three sets, but their interpretation changes according the scheme used.

This is performed on every single measurement, the only purpose of the pre-processing phase is to select the integration time windows that split measurement sessions. No individual lidar profile is discarded in the process. Of course, measurements that were flagged as clear sky situations in the pre processing will most likely not result in any scattering layer identification as the optical parameters will not overcome the background thresholds. However, since the pre-processing phase is based on an approximated inversion, we consider it safer not to exclude them from the layer detection procedure.

Above statements are now included as edition in lines **188-194**: "Before the inversion procedure, lidar data are first pre-processed to adequately set the integration window of the individual 3 minutes raw files ensuring homogeneity of the lidar scene, either being dominated by clear-air or aerosol/cloud presence. To do so, a preliminary inversion assuming no particulate extinction is performed on fifteen minutes blocks to derive a non-corrected for extinction (nce) scattering ratio. A clear sky or Aerosol/cloud presence tag is applied to this preliminary product. Then, the 3-minutes raw files are summed accordingly. This pre-processing avoids the spatio-temporal smoothing

necessarily induced by blending in clear-air before or after any cloud detection and leads to a better Signal to Noise Ratio (SNR)."

50) Line 144: "assuming no particular extinction" – do you mean "assuming no particulate extinction"?
Yes, mistake corrected. Edited "no particulate extinction" line **190**

51) Lines 155-169: As mentioned above, I suggest you move these lidar parameter definitions up to Section 2.1
This point was addressed in comment 33).

52) Section 3.2 Classification schemes: General comment- I think too much emphasis has been put on discussion and evaluation of the classification schemes. You state that the purpose of the study is not to "review, rank or assess" the classification schemes, but to "find a proper framework" to analyse your data. It sure seems that you are reviewing and assessing the classifications. What does "proper framework" mean? How do you decide which classification scheme provides the proper framework.
The point is actually close to the goal of the paper, and the introduction has been reworked to make it clearer ; besides, the sentence pointed out by the reviewer is indeed misleading. We acknowledge that the location of DDU and the associated dataset may not be be representative of the whole antarctic continent and therefore not suited to validate the full PSC spectrum covered by the classification schemes, especially regarding the larger ENH and ICE particles. In turn, we find it meaningfull to consider different schemes built on different optical properties to enrich the analysis our data. The paper aims at providing the community with a PSC distribution above the station. The consistency of this distribution may only be assessed from different schemes, and this work remains to our knowledge rarely made, the most recent one being Achtert et al. (2014) and applied to the Arctic.

The sentence pointed out by the reviewer was modified to present the purpose of this section more clearly, edited lines **228-230**: "The purpose of this section is notably to provide the community with a PSC distribution which consistency has been checked using different observational parameters and thresholds. Applying different classification schemes emphasizes the variability and in turn, consistency, of our distribution."

53) Lines 176-182: Much of this was already discussed in the Introduction.
It is indeed repetitive, and was replaced by the following sentence, lines **230-232**: "As mentioned in introduction, the classifications B05 and P11 are considered here following the conclusion of Achtert et al (2014), to which we had P18, the update of P11 published in 2018."

54) Line 225: "10 2 order or magnitude" should be "10 2 order of magnitude" Do you really mean 100 orders of magnitude or just 2 orders of magnitude (factor of 100 in magnitude)?
It is a mistake, we are indeed talking about 2 orders of magnitude. Edited "two orders of magnitude" line **276**

55) Lines 230-232: I agree that it doesn't make sense to use the MLS measurements directly. But wouldn't be better to use a climatology of HNO3 and H2O and have a time dependent threshold? It would be straight forward to produce a climatology from the MLS data.
Following suggestions of all reviewers, a comparison of CALIOP data in the DDU vicinity and DDU lidar was added to the study. Therefore, we extracted the $R_{T,ICE}$ threshold values used by

CALIOP around DDU and decided to use them in our analysis fueled by the P18 classification. The manuscript was edited accordingly, lines **281-283**:

"We choose to use the $R_{T,ICE}$ values provided by CALIOP in the PSC Mask v2 product. For each DDU lidar measurement, the $R_{T,ICE}$ value was taken from the closest CALIOP profile in both time and space, within the 100 km radius area centered on the station."

56) Line 235: Suggest rewording this to read: "The wave ice category defined in P11 and P18 was ignored in this study ..."
Edited as suggested: "The wave ice category defined in P11 and P18" line **284**

57) Line 236: verb tense doesn't match- "... published classifications ... features" should be "...published classifications ... feature"
Typo corrected line **287**, "feature".

58) Line 242: Suggest changing "disequilibrated" to "non-equilibrium"
Corrected, "non-equilibrium" line **292**

59) Discussion of Figure 2: Figure 2d (derived from Tesche et al., 2021) is based on only two Antarctic seasons (2012 and 2015) and I believe uses only a subset of CALIOP measurements randomly selected to represent the possible sampling of a ground-based lidar that is affected by cloudiness and other measurement-inhibiting factors. What is the relevance of this figure to the others in Figure 2 that are based on 14 years of data? Doesn't seem to be a fair comparison. It certainly would be straight forward to derive a new figure using the CALIOP data for the same timeframe as your DDU data- then the comparison would be more meaningful.
We thank the reviewer for noting this mistake. Reading Tesche et al. (2021), we actually missed that the Antarctic comparison is only based on 2012 and 2015 while the Arctic one is based on all winters from 2006 to 2018. To address this important mistake, we extracted the CALIOP measurements around DDU from 2007 to 2020 and produced a new figure and edited the analysis accordingly.

To properly account for this new content, a section 2.2 was added lines **140-147** to introduce CALIOP. A section 3.2, lines **202-223**, was also added to present the PSC detection method of CALIOP compared to the one we use. A paragraph was added in the beginning of section 4.1., lines **296-309**, to detail the statistics of CALIOP overpasses above DDU. The discussion of figure 3 was adapted adequately.

In the reviewed PSC distribution, we count the PSC layers for each type. Introducing a comparison with CALIOP measurements leads us to adapt our counting approach. Since CALIOP sorts each vertical bin separately (with a coherence criterium), it takes the geometrical thickness of PSC into account. This is relevant and we modified our method accordingly. Therefore, we now take into account each vertical bin of the identified PSC layer as explained in section 3.2. This explains the change in the distribution in figure 3 as compared to the reviewed version.

Here is the new version of Figure 3:

[Figure]

Figure 3: PSC types distribution observed at DDU for the three considered classifications: B05 (a), P18 (b), P11 (c) and observed by CALIOP extracted around DDU using P18 scheme (d).

Edited lines **198**: "... is attributed to each vertical bin of the layer."

60) Discussion of ice discrepancy in Lines 271-274: To better investigate this- I suggest you subset the CALIOP data to the DDU location and evaluate the ICE abundance on days when the DDU lidar operated versus days in which DDU lidar didn't operate.
We indeed extracted CALIOP measurements at DDU as explained before. Doing so, we could identify that ICE PSC have only been detected by CALIOP at DDU on 19 different days between 2007 and 2020. This sample seems too low to build statistics. We can however say that DDU lidar was rarely operated on these days (only 5 out of these 19 days), suggesting a possible important tropospheric cover or bad weather condition hindering operations. We now mention this in the manuscript.

Edited lines **335-338**: "Between 2007 and 2020, CALIOP detected ICE PSC above DDU on 19 different days, out of which 4 correspond to DDU measurements, suggesting a possible important tropospheric cover or bad weather condition hindering operations. However, we do not consider this small sample robust enough to support the analysis."

61) Lines 280-283: These two sentences are not clear- not sure what you're trying to say.
Our point here considers the depolarization threshold used in B05 (10%) along with the few NAT clouds identified at DDU using this threshold: we consider the latter number to be too conservative given the expected abundances of NAT clouds. As compared to other classifications which include "NAT mixtures" (MIX1 and MIX2 in P11 and NATmix in P18), the definition of NAT may be more restrictive in B05. This is what was implied by "pure NAT clouds": we can consider that the NAT

clouds identified with B05 are PSCs entirely composed of – or at least highly dominated by - NAT particles, and not mixed with significant amounts of STS droplets.

This was edited to make it clearer, lines **360-363**:

"Given the 10% depolarization threshold and the relatively low amount of NAT clouds identified by B05, we consider that B05 classifies as NAT the PSC that are only composed of, or highly dominated by NAT particles. Whereas P11 and P18 separate NAT mixtures into MIX1, MIX2 and NATmix which may include a significant share of STS droplets."

62) Lines 285-286: Why would optical properties be dependent on latitude?

Sentence actually needs to be rephrased ; we do not imply some direct correlation between latitude and optical properties, but rather that some range of optical properties need thermodynamical conditions rarely met at lower antarctic latitudes : ICE PSC persistence is related to temperature remaining below $T_{ICE}$, and out point is simply to state that optically equilibrated persistent ICE PSC layers are rarer above DDU than above stations at higher latitudes.

Edited in the manuscript to be clearer, lines **365-367**:

"The relative low number of ICE events we report relates to the fact ICE PSC fields above DDU are more unstable that those remaining deep inside the vortex. The optical properties of the ICE clouds observed at DDU are thus expected to be closer ...."

63) Line 290: "barely never" – that phrase makes no sense. Do you mean "barely ever"? Probably would be better to just say "rarely detected."

Corrected as suggested "rarely" line **370**

64) Figure 4 and corresponding discussion: Using threshold temperature values calculated with fixed values of HNO3 (10 ppbv) and H2O (5 ppmv) will likely produce misleading results. These values may be appropriate for early season (at ~50 hPa), but clearly are not representative for the bulk of the season after denitrification and dehydration have occurred. In reality, the gas phase abundances are much lower over most of the season and the threshold temperatures will correspondingly be lower. It would not be too difficult to derive a climatology of HNO3 and H2O from MLS data that reflects the seasonal and altitude variation and then use this to calculate time dependent temperature thresholds. This would provide a much more realistic evaluation of the PSC detections versus altitude and temperature. But then I ask, is the analysis presented in Figure 4 even necessary for this paper? What is the purpose of this analysis?

Figure 5 illustrates several points that sound consistent to us. First, it shows the altitude distribution of each PSC type. It offers another point of view of the different distributions produced by the three classifications shown in Figure 3.

Considering this comment and the following concerning PSC formation temperature thresholds, we indeed made the approximation of early winter conditions. The idea was not to use MLS measurements at DDU, just like the $R_{T,ICE}$ threshold in P18. Following the reviewer's comments we now do calculate $T_{NAT}$, $T_{STS}$ and $T_{ICE}$ using the closest MLS $H_2O$ (and $HNO_3$ when necessary) measurement available. To include this change in figure 5, we needed to change its structure : the figure now presents the temperature relative to the relevant threshold ($T_{NAT}$, $T_{STS}$ or $T_{ICE}$) depending on the type considered. For each PSC measurement, we calculated the threshold temperature corresponding to the daily abundance of $H_2O$ and $HNO_3$. Still, this approach doesn't account for the kinetics of PSC formation, and embeds the relatively large uncertainties of MLS measurements at PSC altitudes. Thus, the temperature shifts observed mainly for the STS class are completely acceptable to us, especially considering STS formation temperature is apart from the NAT and ICE one, as it is not associated to a discrete physical phase transition but to a continuous chemical composition change within the droplet.

The analysis of Figure 5 has been adapted to the new layout. The new version of Figure 5 is shown below:

[Figure]

Figure 5: Distribution of PSC detection at DDU from 2007 to 2020 as a function of altitude and temperature relative to the relevant threshold, $T_{STS}$, $T_{NAT}$ or $T_{ICE}$. Lines correspond to STS, NAT + mixtures and ICE from top to bottom. Columns correspond to the classification schemes B05, P11 and P18 from left to right.

Lines **380-381**: "For each PSC measurement, MLS $H_2O$ and $HNO_3$ concentrations are used to compute the relevant threshold temperatures."

Lines **382-383**: Figure 5 presents the distribution of PSC measurements as a function of altitude and temperature relative to the relevant type threshold."

Lines **384-389**: "... Temperature is not a variable in our PSC detection method, yet we note that most NAT+mix measurements are below the $T_{NAT}$ threshold within expected uncertainties related to MLS and ERA5 data. Considering STS, it appears that temperatures are above the threshold for all schemes. This is partly expected as $T_{STS}$ is not associated to a discrete physical phase transition but to a continuous chemical composition change within the droplet. Finally Fig. 5 highlights again that the major difference among classification schemes concerns the ICE category as the three patterns of Fig. 5g, h and i show very different shapes."

Lines **390-391**: "It appears however that P11 classifies few PSC as ICE but those are in the adequate temperature range while B05 and P18 sort most of the ICE PSC above $T_{ICE}$."

Lines **398-399**: "Also, the time of lidar PSC measurement does not exactly match ERA5 data and MLS $H_2O$ and $HNO_3$ measurements which necessarily generates uncertainty."

65) Section 4.2: General comment- why only show one sample PSC event? Now that you have selected a classification scheme- why not process all 14 years of data and produce

statistics on interesting aspects of the PSCs such as the mesoscale characteristics? One example is OK- but how representative is it? A statistical analysis would be very interesting and much more compelling for inclusion in the paper. Can you do this?

This example illustrates some features of PSC fields and bring some fine small-scale considerations as compared to the global point of view of classification schemes.

For example, it highlights the stratified layering mentioned earlier in the paper and in the review. It also highlights the variability as compared to the fixed thresholds of PSC classifications.

This section is not included to be representative of 14 years of measurements, it is here to illustrate several points mentioned earlier in the article, and also aims at showing that groundbased lidars have opportunities to access to the small scale structure and dynamics of PSC fields.

A statistical analysis could be made on the interannual variability, which we did by providing the annual variability in the trend analysis. A interannual variability of the PSC type distribution would use smaller samples not robust enough to produce reliable statistics. Moreover, the low amount of CALIOP overpasses above DDU would further harm the statistics. The current version of the paper provides statistics on CALIOP PSC observation above DDU, highlighting the relatively low PSC occurences at DDU as compared to higher latitude stations.

Even if we don't consider yearly PSC type distribution, the interannual variability of PSC occurences is displayed in Figure 9 as red triangles marks. Besides, the new version of Figure 9 now includes the number of CALIOP PSC days at DDU, merely above 10 per year.

66) Line 320: Figure A2- I think you are actually referring to Figure A1 here. Why put the model analysis in an appendix? I think it is OK to include in the main text.

The mistake in figure numbering was corrected. Edited "Figure A1" line **406**.

We found interesting to mention the agreement with Reprobus for this observation. However, since we did not discuss further this model, the figure was only included in annex. To us, there is no sufficient analysis supporting putting this plot in the core of the paper, especially considering the limitation of PSC modelling in REPROBUS. It is purely presented as mesoscale context.

67) Line 324: Again think you mean Figure A1

Again corrected, line **410** "Figure A1".

68) Line 325: "fully validate the model" is a strong claim- maybe "The model produces PSC at the 435 K and 475 K levels, and no PSC at the 550 K level above DDU, consistent with the lidar measurements."

The statement was indeed too strong. It was edited accordingly, lines **411-412**: "The model produces PSC presence at the 435 K and 475 K levels, and no PSC at the 550 K level above DDU, which is consistent with the lidar measurements"

69) Line 332: "... temperature history cannot be accounted for." I don't think there is a limitation that temperature history cannot be used and in fact future schemes may indeed include temperature history as a parameter. Therefore, I suggest rewording as "... temperature history has not been accounted for."

Rephrased in a safer manner as suggested, line **419**: "has not been accounted for".

70) Line 334: RT and RICE should be written as $R_T$ and $R_{ICE}$

The indices have been corrected, line **420-421**.

71) Figure 5 discussion: Are the $T_{ICE}$ thresholds based on 5 ppmv $H_2O$? This is not likely representative of 28 August when the stratosphere may be severely dehydrated. How would this change your interpretation?

$T_{ICE}$ for this plot has indeed been computed for 5 ppm of water vapor. We checked the water vapour mixing ratio for this day. The measurements from MLS v4.2 and v5 (see the figure attached) show different values at the relevant pressure levels. Since the MLS v4 is still widely used in recent studies, we chose to rely on this version for now.

[Figure]

If this changes the value of the temperature threshold, it would anyway not significantly alter the discussion of Figure 6 since the ERA5 temperature is still slightly below $T_{ICE}$ as well as the discussion of Figure 7 concerning the radiosonde drift and the temperature values at the cloud altitudes.

We still chose to update the figure by computing the $T_{ICE}$ threshold corresponding to these water vapour values, in both figures 6 and 7.

In the initial version of the figure, the temperature clearly dropped below $T_{ICE}$. Taking the stratospheric deshydration into account, the new $T_{ICE}$ computation results in lower temperature threshold values. ERA5 temperatures are now closer to $T_{ICE}$, but still compatible with an ICE PSC formation. The discussion was edited to take this change into account, line 426: "… is slightly above $T_{ICE}$ but then reaches this threshold ..."

72) Figure 5: hard to see the temperature differentials in the bottom panel- would be helpful to reduce the range of the color bar.
The colorbar has been reduced to the -5 / +5 K range in order to better visualize the temperature variations. Figure 6 was also modified to address the comment of another reviewer: the smoothing applied was stronger than mentioned in the text so it was decreased to match the manuscript. Lidar data are smoothed on a 30 min window.

Here is the new version of Figure 6:

[Figure]

Figure 6: 532 nm backscatter ratio of lidar measurements obtained at DDU on the 2015/08/28 (a). The corresponding PSC types according to P18 classification scheme (b) and ERA5 temperatures at DDU as compared to the ICE formation threshold $T_{ICE}$, calculated from the MLS $H_2O$ of the day (c). The red dashed line indicates the dynamic tropopause computed from ERA5 data.

73) Figure 7 discussion: I assume the temperature differences are based on the reanalyses data from over DDU (interpolated) and the temperature at the true location of the radiosonde downwind from DDU- right?
They are indeed based on the difference between NCEP, ERA5 and IASI at the DDU location and the radiosonde at its true location. In other words, the radiosonde drift has not been taken in account here. We think the comparison is still relevant and we extensively detailed this choice in our response to the comment 79.

74) Line 379: "NCEP is obviously less accurate ..." Is it obvious? Differences with the radiosonde are larger- but doesn't necessarily imply NCEP is wrong. Do you have a citation that NCEP is less accurate than ERA5 and IASI?
The issue on the temperature bias caused by the radiosonde drift is extensively discussed later on in comment 79.
As for the model intercomparison, we do not claim the ERA5 products are better than the latest NCEP reanalyses. Historically, ECMWF stratospheric meteorological products tended to be a bit more accurate than NCEP products (see references below), notably for products relevant to our study (i.e. stratospheric Antarctic temperatures).
However, despite the multiple publications from the SPARC Reanalysis Intercomparison Project (S-RIP) (such as the special issue on this topic in Earth System Science data: https://essd.copernicus.org/articles/special_issue10_829.html), it is not very obvious that the latest ECMWF reanalyses, ERA5, still have an edge on the NCEP products.

When compared to the high vertical resolution Global Navigation Satellite System radio occultation (GNSS RO) data, ERA5 shows obvious improvements in temperature data compared to ERA-I and also a slightly better agreement with GNSS RO measurements than MERRA2. For the 2007–2017 period, ERA5 shows the best agreement with observations, while the other two reanalyses (ERA-I, MERRA2) slightly warm biased. Clearly, ERA5 reanalyses represent a substantial improvement on ERA-I reanalyses which were already widely used (Hoffman et al., 2019).
We have not found a single publication demonstrating the better quality of the NCEP products compared to ERA5.

https://rmets-onlinelibrary-wiley-com.insu.bib.cnrs.fr/doi/abs/10.1256/qj.03.76
Nevertheless, it is found that NCEP/NCAR reanalyses tend to be slightly warmer (0.8 K) than the observations, while the converse is true for ECMWF analyses (−0.3 K). Finally, trajectory comparisons are performed. It is found that trajectories built with ECMWF winds are more accurate than those built with NCEP/NCAR winds.

https://agupubs-onlinelibrary-wiley-com.insu.bib.cnrs.fr/doi/full/10.1029/2008JD010116
This article compares the temperature, zonal, and meridional velocities issued by the 50-year National Centers for Environmental Prediction–National Center for Atmospheric Research (NCEP-NCAR) Reanalysis (NN50) and European Centre for Medium-range Weather Forecasts (ECMWF) operational analyses with independent observations collected during the Vorcore superpressure balloon campaign. The ECMWF analyses are found to be more accurate than the NN50 reanalyses. In particular, an overall warm bias in the polar Southern Hemisphere lower stratosphere is found in NN50 (+1.51 K), while a cold bias is found using ECMWF analyses (−0.42 K).

https://journals-ametsoc-org.insu.bib.cnrs.fr/downloadpdf/journals/mwre/134/11/mwr3256.1.pdf
The results of these comparisons indicate that NN50 tends to be a few degrees colder than the observations in the SH subpolar latitudes, while ERA-40 is less hit by this cold-pole issue

https://agupubs-onlinelibrary-wiley-com.insu.bib.cnrs.fr/doi/full/10.1029/2001JD001329
In 2000 the standard deviations of ECMWF, MO, and DAO with respect to the measured temperatures range from 1.0 to 1.3 K, whereas NCEP and REA have substantially larger errors. In 1999 the flights took place during a major warming, and all operational models had large standard deviations and substantial biases. Preoperational versions of the new ECMWF model with increased stratospheric resolution and assimilation of the advanced microwave sounding unit, which none of the other models assimilated, show small biases and standard deviations.

75) Lines 388: This statement should appear in the Introduction and help define the focus of the paper!
This statement was moved in introduction as suggested and rephrased accordingly in this section in order to avoid repetition.

Moved to introduction, lines **82-84**: "Two of the major roles of a ground station are to perform process studies and establish decadal trends. Such trends are highly valuable because they reflect the evolution of the stratosphere, in terms of temperature and chemical compositions. "

Edited line **475**: "As mentioned in introduction, ground stations are key to the establishment of decadal trends."

76) Line 395: "Both NCEP and ERA5 ..." You concluded in the previous section that the NCEP temperatures are not as accurate as the ERA5 and therefore the ERA5 data would be used in this study. Why include the NCEP data here?
As discussed in a previous comment (74), NCEP is considered less accurate at DDU than ERA5. It still does not make the comparison proposed in Figure 9 useless. We are not using NCEP instead of ERA5, we are using both. And we modified Figure 9 to also include IASI as this was suggested by another reviewer. The inclusion of NCEP here adds valuable information in that, despite the discrepancies between datasets highlighted by figure 8, the two model and the spaceborne observationally derived trends are globally consistent.

77) Lines 399-406: How was T NAT calculated for the trend analyses? Did you use a fixed value of 10 ppbv over all altitudes and days? Would a more accurate value reflecting denitrification change your results?
The original version of the figure indeed included a constant $T_{NAT}$ value calculated from a 10 ppb value for $HNO_3$ and 5 ppm vor $H_2O$. Daily MLS measurements were extracted around DDU and the $T_{NAT}$ values are now calculated based on these time dependent values. The trend and the corresponding figure have therefore been changed. The trend shifts to -5.7 to -4.6 PSC days per year per decade without significant change on the analysis.
It is important to note that taking the stratospheric denitrification into account tends to decrease the computed $T_{NAT}$ values. As an expected result, the ΔT criteria adjusted to our lidar measurements is now -1 K and not -2 K as in the initial version of the manuscript. Figure 9 and the associated discussion have been edited on lines **490** and **492** and in the caption of Figure 9.

78) Trend analyses: Did you consider subsetting the CALIOP PSC data from 2007-2020 to the DDU location and compare this with the temperature time series? This would be a very interesting exercise to include here.
Now that we have extracted CALIOP PSC Mask v2 product around DDU, we can include such a comparison. As it was highlighted in the results of the comparison of CALIOP and DDU lidar

measurements, there are few CALIOP profiles featuring PSC detection available at DDU, so that computing a trend per year is not reliable. The number of days with CALIOP PSC detection at DDU, per year, is now featured in figure 9 as black triangles. We find on average 6 PSC days per year detected by CALIOP around DDU, with a minimum of 1 (2012, 2014) and a maximum of 11 days (2009). We note that this would produce a slightly negative trend of -1.1 PSC days per decade, but we consider that building a trend on so few PSC detections is not significative as discussed in a previous comment.

79) Line 435: "... ERA5 slightly overestimates ..." This conclusion is based on comparisons with radiosondes not necessarily collocated with DDU due to balloon drift. Doesn't the drift of the balloon make it difficult to conclude anything quantitative about accuracy since there are spatial temperature gradients that introduce differences?
First, let us state on the radiosonde drift: radiosonde temperature profiles are not used in the lidar inversion because of the low burst altitude in winter. We mostly use them to further investigate on borderline situations such as discussing the type of a PSC detected or the temperature related to an aerosol layer.

When doing so, it is important to keep in mind that the radiosondes can experience significant drift, especially in winter, and therefore not reflect the temperature of the atmospheric column sounded by the lidar. However, this does not mean that these radiosondes are not globally representative of the climate aroud DDU and that they cannot be used to evaluate the relevance of temperature datasets at DDU, the drift remaining below the 1° pixel resolution in model grids. The plot below shows the mean distance of the radiosondes to DDU for each 1 km interval, for the whole year, summer and winter.

There are two main reasons why we consider that the comparison of the temperature datasets with the radiosondes is relevant despite the point we made about their drift.

The location of DDU is 66.6°S – 140°E. IASI extraction area around DDU is set to 66.8°S – 66.5°S and 139.7°E – 140.3°E, which correspond to approximately a rectangle of 26,5 x 33,4 km.

The mesh of the ERA5 product used in our study is 0.25° x 0.25°, which is an area of approximately 27.8 x 11.1 km at DDU.

NCEP provides temperature and altitude profiles for DDU station, at the location 67°S – 140°E. The real location of the station being 66,66°S – 140°E, NCEP interpolation is approximately 37,8 km from the true lidar location.

Therefore, considering that up to an altitude of 15 km the radiosondes are on average closer than 50 km of DDU (see figure attached), and considering the distances at stakes for the extraction of IASI measurements or the sizes of NCEP and ERA5 meshes, it is reasonable to say that these datasets are representative of the same area. When comparing NCEP, ERA5 and IASI to these radiosondes as shown on figure 7, even when only focusing on altitudes below 15 km, it appears than NCEP is less accurate than ERA5 and IASI at DDU.

[Figure]

The other reason is more pratical. In order to account for the drift of the radiosondes in figure 8, we would need to adjust NCEP, ERA5 and IASI at the true location of the radiosonde every day. While it is realistic with the 4xdaily ERA5 and its 0.25°x0.25° grid, it would raise issues for IASI and NCEP. First, the NCEP product used in this study is provided already interpolated at DDU location, so we would need to use another product to access to temperatures around DDU.

Depending on the location of DDU in the field of view of IASI, taking the drift of the radiosonde in account would sometimes imply to produce a temperature profile composed of different orbits and therefore different times of measurement, and then compare it the radiosonde. This would make no sense and would in many cases be worse than ignoring the drift of the radiosonde in the comparisons.

Finally, NCEP interpolation at NDACC station is provided daily, while the ERA5 product we interpolate at DDU is available 4xdaily, which is of course an advantage to be as close as possible to the lidar measurements.

80) Figure 9: I find the figure confusing- but maybe I just don't understand what is being shown. For PSC thickness between about 2-7 km, the T<T NAT domain is smaller than the PSC thickness? But you state that the figure shows that the T<T NAT domain is significantly larger than the actual PSC thickness! What am I missing?
The discussion of this figure may not be clear enough and related to the way the figure is built. For each PSC detection at DDU, we computed the total stratospheric domain occupied by PSC layers, i.e. total PSC thickness and we also computed the height of the stratospheric domain satisfying $T < T_{NAT}$ on this very day. Then, the distribution of both values computed for the whole dataset were plotted. So the highlight of the figure is that, while PSC thickness is approximatively equally distributed between 1 and 7 km, the stratospheric domain under $T_{NAT}$ when PSC are detected is significantly higher, often larger than 10 km. The point is to show that PSC layers do not fill the whole stratospheric domain satisfying $T < T_{NAT}$, because PSC volume were sometimes evaluated this way.

We consider that this figure provides interesting information on the thickness of PSC layers as compared to their potential "domain of existence".

Since this figure was not clearly understood by two reviewers, we adapted it to be more straight forward. For each day with a PSC detection, we calculated the stratospheric range satisfying the condition $T < T_{NAT}$. $T_{NAT}$ was calculated with the daily MLS $H_2O$ and $HNO_3$ measurements. Let us call this range $H_{NAT}$, expressed in km. We also calculated the geometrical thickness of the PSC detected, called $H_{PSC}$. Then, we computed the difference $H_{NAT} - H_{PSC}$: it represents the stratospheric range satisfying $T < T_{NAT}$ unoccupied by PSC layers. The distribution of this difference, for all PSC detections at DDU from 2007 to 2020 is plotted in Figure 10. It shows that PSC do not occupy all the stratospheric volume satisfying $T < T_{NAT}$ so that estimating the first quantity by computing the second is not a satisfying assumption. On this figure are only represented the PSC measurements for which ERA5 temperatures dropped below $T_{NAT}$ otherwise it would have featured negative values. Such cases represent 29% of all PSC detection, they can be caused by the approximation in the calculation of $T_{NAT}$, the uncertainty of ERA5 or an outlier in our PSC detection method.

Here is the new version of Figure 10:

[Figure]

Figure 10: Distribution of the thickness of the stratospheric domain satisfying T<T$_{NAT}$ unoccupied by PSC layers, in km, on days when a PSC is detected at DDU.

81) Section 5 Conclusions: No specific issues at this point. Will reserve comment for the anticipated revised version.

References:

Ansmann, A., Ohneiser, K., Chudnovsky, A., Knopf, D. A., Eloranta, E. W., Villanueva, D., Seifert, P., Radenz, M., Barja, B., Zamorano, F., Jimenez, C., Engelmann, R., Baars, H., Griesche, H., Hofer, J., Althausen, D., and Wandinger, U.: Ozone depletion in the Arctic and Antarctic stratosphere induced by wildfire smoke, Atmospheric Chemistry and Physics, 22, 11 701–11 726, https://doi.org/10.5194/acp-22-11701-2022, 2022.

Rieger, L. A., Randel, W. J., Bourassa, A. E., and Solomon, S.: Stratospheric Temperature and Ozone Anomalies Associated With the 2020 Australian New Year Fires, Geophysical Research Letters, 48, e2021GL095 898, https://doi.org/https://doi.org/10.1029/2021GL095898, e2021GL095898 2021GL095898, 2021.

Snels, M., Scoccione, A., Di Liberto, L., Colao, F., Pitts, M., Poole, L., Deshler, T., Cairo, F., Cagnazzo, C., and Fierli, F.: Comparison of Antarctic polar stratospheric cloud observations by ground-based and space-borne lidar and relevance for chemistry–climate models, Atmospheric Chemistry and Physics, 19, 955–972, https://doi.org/10.5194/acp-19-955-2019, 2019.

Snels, M., Colao, F., Cairo, F., Shuli, I., Scoccione, A., De Muro, M., Pitts, M., Poole, L., and Di Liberto, L.: Quasi-coincident observations of polar stratospheric clouds by ground-based lidar and CALIOP at Concordia (Dome C, Antarctica) from 2014 to 2018, Atmospheric Chemistry and Physics, 21, 2165–2178, https://doi.org/10.5194/acp-21-2165-2021, 2021.

Stone, K. A., Solomon, S., Kinnison, D. E., and Mills, M. J.: On Recent Large Antarctic Ozone Holes and Ozone Recovery Metrics, Geophysical Research Letters, 48, e2021GL095 232,

https://doi.org/https://doi.org/10.1029/2021GL095232, e2021GL095232 2021GL095232, 2021.

---

## Author Comment (AC3)

We thank the reviewer for their comments. Below is our point by point response in blue.

We draw the reviewer's attention to the fact that, following a similar comment of all reviewers, a figure 1 has been added to the manuscript, featuring information on the operation statistics of DDU lidar. Therefore, all figure numbers have been incremented accordingly as compared to the first version of the manuscrit.

Also, to address another comment, the threshold temperatures  $T_{NAT}$ ,  $T_{STS}$  and  $T_{ICE}$  are now computed based on the closest MLS H2O and HNO3 concentration measurement. Although this does not change their meaning or their interpretation, most of the figures have gone through some slight changes.

**General / major comments:**

G1) The lidar at Dumont d'Urville is in operation since 1989, why do you only focus on the last 14 year? Did the occurrence of PSCs in general and of different PSC types changed since 1989? Do you observe a trend since 1989?
Even if the original setup of the lidar dates back to 1989, it has not been in continuous operation since then. Still, it is continuously operating since 2006 with the same laser source installed in 2005. The monitoring calendar has also been greatly enhanced from 2007 onwards, providing a consistent time series motivating the choice of this time period, also coincident with the launch of the CALIPSO mission. As a whole, spaceborne coverage and recent version of the products are also actually better on this time frame. In the 1989-2006 time period, the monitoring policy was different partly due to the presence of a colocated ozone lidar. As mentionned in the paper line 513-514, no significant PSC trend was established in the 1989-2008 time interval from David et al., (2010)". The manuscript has been edited accordingly, line 80: "Considering the latest laser source replacement in 2005 and the continuous monitoring from 2006, ..."

G2) It would be good if the authors could add a paragraph about the lidar measurements including some statistics, e.g. How many days per year the lidar was operational? How many PSC were observed per year? When were there observed? Are the PSC observation evenly distributed over the winter?...

Following the comment of another reviewer we included statistics on the operation of DDU lidar. The newly aded figure 1 presents the number of measurement days per year, from 2007 to 2020 as well as the duration of these measurements per month, in average. The number of PSC days observed per year is represented by the red triangles of figure 9. The temporal distribution of PSC occurences during the winter season does not support the core of the analysis around the PSC type distribution using different classification schemes, and we choose not to include it in this study. Here is the new Figure 1:

Figure 1: Operation statistics of DDU lidar. (a) Number of measurement days per year, from 2007 and to 2020 and (b) mean duration of measurement sessions per month, in minutes, from 2007 to 2020.

G3) Please include a more detail analysis of the 14-year dataset: e.g. year to year variability, comparison to other ground-based stations and CALIPSO

We address here and below the comments on year-to-year variability as a whole, and comparison to other groundbased and spaceborne datasets:

A statistical analysis on the interannual variability is provided to an extent in the trend analysis, which precisely includes features the statistics asked by the reviewer: year-to-year PSC variability at DDU. A interannual variability of the PSC type distribution would use smaller samples not robust enough to produce reliable statistics so we choose not include it.

Other reviewers mentioned the need to compare with other groundbased datasets. It is now included between lines **341** and **354** where the types distribution we observe at DDU is compared to those observed at McMurdo (2006-2010) and Concordia (2014-2018) by Snels et al., (2019 and 2021) respectively.

As mentioned in our response to the other reviewers, we did a mistake when reading Tesche et al. (2021) and did not realize that while all winters from 2006 to 2018 are considered for the Arctic stations, only 2012 and 2015 are used for Antarctica. To address this issue, we extracted the CALIOP PSC Mask v2 product around DDU and included it in the study. To do so, we used a method pretty similar to the one described by Snels et al. (2021) when comparing groundbased measurements from Concordia with CALIOP measurements.

In the reviewed PSC distribution, we count the PSC layers for each type. Introducing a comparison with CALIOP measurements led us to adapt our counting approach. Since CALIOP sorts each vertical bin separately (with a coherence criterium), it takes the geometrical thickness of PSC into account. This is relevant and we modified our method accordingly. Therefore, we now take into account each vertical bin of the identified PSC layer as explained in section 3.2. This explains the change in the distribution in figure 3 since this reviewed version.

To include this new content in the mansucript, a presentation of CALIOP was included at lines **140-147**. A presentation of the method of PSC detection with CALIOP is included at lines **202-223**. Finally, the types distribution produced for DDU for 2012 and 2015 was replaced in Figure 3d by the CALIOP data extraction and the subsequent discussion has been edited (lines **296-309**, **311-312** and **335-338**).

Even if we don't consider yearly PSC types distribution, the interannual variability of PSC occurences is displayed in Figure 9 as red triangles marks. Besides, the new version of Figure 9 now includes the number of CALIOP PSC days at DDU, merely above 10 per year.

**Here is the new version of Figure 9:**

Figure 9: PSC days per year at DDU from 2007 to 2020 featuring PSC detection with the lidar in red triangles. Potential PSC days per year estimated by ERA5, NCEP and IASI based on the lidar measurements are shown in green and red respectively. Green, blue and fuchsia lines represent the corresponding trends. Grey arrows indicate the number of days per year where the T -  $T_{NAT}$

---

## Referee Report (RR1)

**Review of Revised Manuscript acp-2022-401-v3**

I would like to thank the authors for their detailed replies to the referee's comments. Overall, the authors have addressed most of my concerns and the revised manuscript is significantly improved over the previous version. I do not have any major concerns with the revised manuscript, but there are a few minor points that should be addressed before final publication.

Minor Comments:

Line 8: CALIOP acronym should be defined in the Abstract.

Line 26: "… variations in account…" should be "…variations into account…"

Line 34: suggestion replacing "…, which role…" with "…, whose role…"

Equations 1-5: Suggest defining all the individual symbols for parallel and perpendicular aerosol/particulate backscatter, etc. Could just include them in the sentences where they are mentioned.

Line 126: "tropopause" do you mean "troposphere"?

Lines 130-131. PSCs may occur above 28 km- would this affect the clear-air reference calculation and calibration?

Line 210: "et" should be "and" ?

Line 222: "As mentioned in introduction…" should be "As mentioned in the introduction…"

Line 336: "… do not directly provides the PSC types distribution…" should be "… does not directly provide the PSC type distribution…"

Figure 5: The distributions shown for P18 over DDU are not consistent with Fig.12 in Pitts et al. (2018) where the 12-year statistics for the entire Antarctic region show STS occurring near $T_{STS}$ and Ice occurring near $T_{ice}$. Can you explain this apparent discrepancy?

Figure 5 caption: Sentence formatting needs fixed.

Figure 9: The black triangles are the CALIOP PSC days over DDU, correct? If yes, then they should be defined in the caption. It is surprising that CALIOP would have such a small number of PSC days. Is this based on a 100-km match distance? Why do you think the number is so small?

---

## Author Response (AR2)

**Author responses to reviewer 1**

Figure 1: add units to y-axis. Add a und b to the Figure.
The figure was updated accordingly.

Can you please add some information on the amount of PSC measurements per year and month, since the focus of the paper is on PSC measurements? It is not enough to provide the PSC days in Figure 9. Could you add the number of lidar observation during the PSC season and the number of PSC observations to the upper plot of Figure 1. And similar in Figure 1b, how often where PSC observed during the different month.
We provide PSC days because delimiting a PSC measurement is arbitrary, not a PSC day. Different methods, such as the one used by Snels et al. (2021) delimit PSCs as vertical bins, which is dependent on the resolution of the lidar design. How can one delimit a PSC in time and space? The only delimitation we consider safe is to say if a PSC was detected during a specific day, this is why we use PSC day.
To account for the reviewer's comment, we added the number of measurements conducted in June, July, August and September as blue stars in the top panel, as well as the total number of PSC days detected in June, July, August and September as green circles in the bottom panel.

Line 195: The used peak detection algorithm needs more explanation. It is not clear how the peak algorithm will work for the particle linear depolarization ratio profile? For example, an STS cloud has a very weak signal in the depolarization. How can a peak algorithm reliable detect such a signal? Also, from Figure 6 it is apparent that your miss to detect parts of the cloud and that the missed part is most likely STS. It would be good if you could provide examples of the you PSC peak detection.
The peak detection algorithm is performed on the total backscatter ratio and on the particle linear depolarization ratio. The results are combined to provide the set of scattering layers. The combination consists of a "logical or", which means that in the case of an STS, the identification of a peak in the $R_T$ signal is sufficient.
As we explained in our first responses to the reviewer, the error in Figure 6 results from a misdetection of the boundaries of the cloud, which can occur with our method. We could have corrected it manually for the purpose of the Figure but we considered it more honest to leave it as it is.
The detection peak algorithm can lead to misidentification of the boundaries of a layer, especially in cases like the one shown in Figure 6 where multiple layers are present. Nevertheless, we consider its efficiency satisfying considering the other options available.

Another point we would like to highlight is that the detection peak algorithm is performed on 15 minutes lidar measurements for the purpose of Figure 6. This is a very short integration period which is not the one used for our climatology: usually we work with measurements integrated on 90 minutes at least. When processing this same measurement integrated on a longer time period, one can see on the figure below that the adequate layers are detected. On the figure below, the layers detected are delimited by horizontal black dashed lines. The red dots indicate the local peaks identified.

[Figure]

Figure 10. The description of the Figure provided in the paper is not sufficient. Can you add the description similar to the one provided in your reviewer answer to the text? Is the x-axis legend correct? If I understand correct thickness means the difference between HNAT – HPSC and the difference can be up to 15km?

"For each day with a PSC detection, we calculated the stratospheric range satisfying the condition T < TNAT. TNAT was calculated with the daily MLS H2O and HNO3 measurements. Let us call this range HNAT, expressed in km. We also calculated the geometrical thickness of the PSC detected, called HPSC. Then, we computed the difference HNAT – HPSC: it represents the stratospheric range satisfying T < TNAT unoccupied by PSC layers. The distribution of this difference, for all PSC detections at DDU from 2007 to 2020 is plotted in Figure 10."

The x-axis legend is indeed correct. This difference can reach values up to 13 km in some rare cases, the 15 km indicated at the right only concern the interpolation of the distribution. Such high values can be reached with a very cold stratosphere and only thin layer of PSC.

To address the reviewer's comment, the discussion of Figure 10 was complemented with the following text, lines 511-526: "To check this above DDU, for each day with a PSC detection, we calculated the stratospheric range satisfying the condition $T - T_{NAT}$. $T_{NAT}$ was calculated with the daily MLS $H_2O$ and $HNO_3$ measurements. Let us call this range $H_{NAT}$, expressed in km. We also calculated the geometrical thickness of the PSC detected, called $H_{PSC}$. Then, we computed the difference $H_{NAT} - H_{PSC}$: it represents the stratospheric range satisfying $T - T_{NAT}$ unoccupied by PSC layers. The distribution of this difference, for all PSC detections at DDU from 2007 to 2020 is plotted in Figure 10."

Now that $H_{NAT} - H_{PSC}$ have been defined in the manuscript, the x-axis label was edited accordingly.

**Author responses to reviewer 2**

Line 8: CALIOP acronym should be defined in the Abstract.

Edited line 8-9: "The Cloud-Aerosol Lidar with Orthogonal Polarization PSC detection"

Line 26: "... variations in account..." should be "...variations into account..."

Edited line 27: "into"

Line 34: suggestion replacing "..., which role..." with "..., whose role..."

Edited line 35: "whose role"

Equations 1-5: Suggest defining all the individual symbols for parallel and perpendicular aerosol/particulate backscatter, etc. Could just include them in the sentences where they are mentioned.

Edited, line 113-115: "… as a function of the parallel and perpendicular molecular backscatter coefficients ($\beta_{mol,//}$ and $\beta_{mol,\perp}$ respectively) and of the parallel and perpendicular particulate backscatter coefficients ($\beta_{aer,//}$ and $\beta_{aer,\perp}$ respectively)"

Line 126: "tropopause" do you mean "troposphere"?

Yes it was a typo and was corrected, line 129. Thank you.

Lines 130-131. PSCs may occur above 28 km- would this affect the clear-air reference calculation and calibration?

If such PSC would occur, it would indeed affect the inversion procedure. Such cases are very rarely met at DDU and when it is the case, the clear-air reference altitude is manually increased.

Line 210: "et" should be "and" ?

It is indeed a typo, corrected line 213: "… and ..."

Line 222: "As mentioned in introduction..." should be "As mentioned in the introduction..."

Edited line 225: "… the introduction"

Line 336: "... do not directly provides the PSC types distribution..." should be "... does not directly provide the PSC type distribution..."

Corrected line 336 as suggested: "... does not directly provide the PSC type distribution..."

Figure 5: The distributions shown for P18 over DDU are not consistent with Fig.12 in Pitts et al. (2018) where the 12-year statistics for the entire Antarctic region show STS occurring near TSTS and Ice occurring near Tice. Can you explain this apparent discrepancy?

Fig. 12 of P18 is based on 12 Antarctic winters for the whole vortex at 21 km. It is based on much more data than our local observations. In the case of CALIOP, the measurements of very cold ICE PSC fields within the core of the vortex probably explain the differences with our observations as they drive the distribution. The ICE PSC detected at DDU are on average observed at higher temperature due to the location of the station, at the edge of the vortex and once again the low number of ICE observations could on its own explain the discrepancy.

As for STS PSC, the formation temperature spans a few Kelvins depending on the HNO3/H2O fraction inside the droplet. In our case, we use the closest MLS profile both in time and space which is less accurate than CALIOP who can take advantage of its synchronicity with MLS. Moreover, as mentioned in the manuscript, we are not surprised by the gap between our observations and $T_{STS}$ given the non-discrete nature of this variable.

Figure 5 caption: Sentence formatting needs fixed.

The formatting issue of the caption was addressed.

Figure 9: The black triangles are the CALIOP PSC days over DDU, correct? If yes, then they should be defined in the caption. It is surprising that CALIOP would have such a small number of PSC days. Is this based on a 100-km match distance? Why do you think the number is so small?

The caption was addressed to mention the black triangles. Edited in the caption: "..., and with CALIOP in black triangles."

Concerning the low number of CALIOP observations at DDU, it is indeed based on a 100-km radius area around the station. First, we expected this low number since Tesche et al. (2021) already mentions a significative difference in the CALIOP coverage between stations like DDU and the ones deeper inside the continent, such as Dome C Concordia.

First, the location of DDU at the edge of the vortex explains partly a lower number of PSC observations as compared to higher latitude sites. As Tesche et al. (2021) shows it, especially with its Figure 8, there is an important gap in profile avaibility between the stations located on the shore of Antarctica and the ones located deeper inside the continent.

---

## Author Response (AR3)

Dear Matthias,

Thank you for your feedbacks and for the processing of our article's reviews.

I adapted the figure sizes and especially the ticks and ticks labels sizes in the updated files. I hope it is more adapted now.

Best regards,
Florent